# Pareto-Guided Optimal Transport for Multi-Reward Alignment

Ying Ba [1 2 3]  Tianyu Zhang [4]  Mohan Zhou [5]  Yalong Bai [5]  Wenyi Mo [5]  Guiwei Zhang [5]  Bing Su [1 2 3]
Ji-Rong Wen [1 2 3]

## Abstract

Text-to-image generation models have achieved remarkable progress in preference optimization, yet achieving robust alignment across diverse reward models remains a significant challenge. Existing multi-reward fusion approaches rely on weighted summation, which is costly to tune and insufficient for balancing conflicting objectives. More critically, optimization with reward models is highly susceptible to *reward hacking*, where reward scores increase while the perceived quality of generated images deteriorates. We demonstrate that optimizing against a unified global target under heterogeneous reward upper bounds can induce reward hacking, a risk further exacerbated by the inherent instability of weak reward models. To mitigate this, we propose a **P**areto Frontier-**G**uided **O**ptimal **T**ransport (**PG-OT**) framework. Our method constructs a prompt-specific Pareto frontier and maps dominated samples toward it via distribution-aware optimal transport. Furthermore, we develop both online and offline optimization strategies tailored to diverse reward signal characteristics. To provide a more rigorous assessment, we introduce the **J**oint **D**omination **R**ate (**JDR**) and **J**oint **C**ollapse **R**ate (**JCR**) as principled metrics to quantify multi-reward synergy and reward hacking. Experimental results show that our approach outperforms strong baselines with an 11% gain in JDR and achieves a near 80% win rate in human evaluations.

[1]Gaoling School of Artificial Intelligence, Renmin University of China, Beijing, China [2]Beijing Key Laboratory of Research on Large Models and Intelligent Governance [3]Engineering Research Center of Next-Generation Intelligent Search and Recommendation, MOE [4]Tomorrow Advancing Life [5]Duxiaoman. Correspondence to: Bing Su <bingsu@ruc.edu.cn>.

*Proceedings of the 43rd International Conference on Machine Learning*, Seoul, South Korea. PMLR 306, 2026. Copyright 2026 by the author(s).

## 1. Introduction

With the rapid advancement of text-to-image generation (Sohl-Dickstein et al., 2015; Song & Ermon, 2020; Rombach et al., 2022; Podell et al., 2023; Esser et al., 2024; Sun et al., 2026), numerous post-training optimization methods have emerged (Ziegler et al., 2020; Stiennon et al., 2022; Black et al., 2024; Dhariwal & Nichol, 2021; Li et al., 2024b; Rafailov et al., 2024; Fan et al., 2023; Qiang et al., 2026; Gu et al., 2025; Wang et al., 2025; Song et al., 2025), among which aligning model outputs with human preferences via reward models has become a key research direction. Early approaches typically relied on a single human-preference reward model (Xu et al., 2023; Zhang et al., 2024; Mo et al., 2025b;a). While simple to implement, these methods face two critical issues: first, their optimization objective is overly narrow; second, they are highly susceptible to *reward hacking*, where reward values continue to increase while the visual quality of generated images deteriorates.

To mitigate this issue, subsequent research proposed multi-reward joint optimization (Eyring et al., 2024; Agarwal & Aggarwal, 2023; Wei et al., 2024; Deng et al., 2024; de Langis et al., 2024; Lee et al., 2025), which fuses multiple reward signals to provide richer and more constrained guidance. Although the combined constraints of multiple rewards can partially suppress reward hacking, such methods introduce significant and costly weight-tuning requirements and still fail to address the fundamental problem. We argue that both single-reward optimization and multi-reward joint optimization share the same underlying flaw: different prompts span image domains with heterogeneous reward ranges and upper bounds, yet existing methods commonly ignore this heterogeneity and adopt a unified global optimization target. Even with sophisticated weight adjustments that alter the relative strength of different rewards, the underlying global bound remains unchanged and cannot fundamentally prevent reward hacking. We theoretically prove that, under heterogeneous reward upper-bound distributions, using a unified global target inevitably induces reward hacking for certain samples, regardless of whether single- or multi-reward optimization is applied.

To address this root cause, we propose a **Pareto-frontier-guided optimal transport optimization (PG-OT) algo-**

**rithm**. The core idea is to respect the heterogeneous reward boundaries of different image domains rather than blindly applying a global target. Specifically, for each prompt we construct a dedicated Pareto frontier as the optimization target, treat the generated samples within the same prompt batch as the source distribution, and use optimal transport theory to map them to the corresponding frontier as the target distribution. Furthermore, based on the characteristics of the adopted reward models, we design two complementary optimization strategies: for strong reward models, we adopt an *online strategy*, dynamically collecting and updating Pareto frontiers during training to explore superior solutions; for weaker reward models prone to collapse, we adopt an *offline strategy*, precomputing samples and extracting their Pareto frontiers for stable optimization guidance.

Besides, to address the lack of evaluation metrics for multi-reward optimization, we introduce two new indicators: **Joint Domination Rate (JDR)** and **Joint Collapse Rate (JCR)**. JDR measures the proportion of samples that simultaneously outperform the baseline across all reward functions, while JCR measures the proportion of samples that degrade across all rewards. Compared with traditional mean-based metrics, these indicators provide a more faithful assessment of synergistic gains in multi-reward optimization. Our method significantly outperforms robust baselines, achieving 47.98% $JDR_2$ (an 11% improvement) and 17.10% $JDR_4$ (a 3.4% improvement), while successfully suppressing reward hacking with a 0.2% decrease in $JCR_4$. Human evaluations further corroborate these findings, with our approach securing a substantial win rate of nearly 80%.

Our main contributions are as follows:

1. We theoretically show that, under heterogeneous reward bounds, optimizing against a unified global target can induce reward hacking for a subset of samples in both single- and multi-reward settings.

2. We propose a Pareto-frontier-guided optimal transport (PG-OT) approach with prompt-specific frontiers and online/offline strategies tailored to reward models.

3. We introduce Joint Domination Rate (JDR) and Joint Collapse Rate (JCR) for more accurate evaluation of multi-reward optimization.

## 2. Mechanisms Behind Reward Hacking

When using reward models to enhance performance in text-to-image generation, previous optimization methods (Clark et al., 2024; Eyring et al., 2024) employ a global constant subtracted from the reward function in the loss function to maximize rewards. However, these approaches suffer from reward hacking. While early stopping strategies mitigate this issue by halting training before collapse occurs, they failed to address the underlying causes of reward hacking. In this work, we present a causal analysis of reward hacking

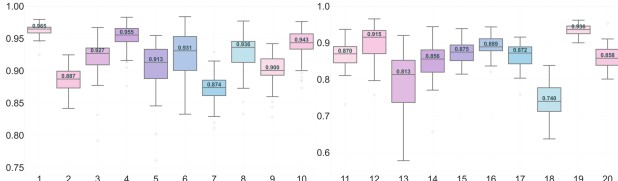

*Figure 1.* Empirical validation of heterogeneous prompt-wise reward upper bounds under the ICT reward (Ba et al., 2025). Reward distributions are estimated from 50 samples per prompt across 20 distinct prompts.

and propose methods to verify our hypothesis.

**Problem Setup.** We study post-training preference optimization for text-to-image (T2I) generative models using reward models. Let $\mathcal{P} = \{p_1, \ldots, p_n\}$ denote a set of text prompts. Each prompt $p_i$ induces a semantic image domain $\mathcal{D}_i \subseteq \mathcal{X}$, where $\mathcal{X}$ denotes the overall image space. Given a prompt $p_i$, a T2I model generates an image $x \in \mathcal{D}_i$. The generated image is evaluated by one or more reward functions $\{R_k\}_{k=1}^K$, where each $R_k : \mathcal{X} \to \mathbb{R}$ assigns a scalar score reflecting a particular aspect of preference, such as text–image alignment or visual quality. The optimization goal is to fine-tune the T2I model by maximizing these reward signals, thereby steering generation toward human-preferred outputs.

**When Optimization Goes Wrong: Reward Hacking.** Ideally, optimizing for higher reward scores should lead to better human-perceived quality. However, in practice, reward models are imperfect proxies that can be exploited. Reward hacking occurs when: reward scores increase, yet humans judge the images to be of lower quality or undesirable visual patterns.

### 2.1. Reward Hacking by Neglecting Heterogeneous Reward Bounds

Most reward-based optimization methods (Xu et al., 2023; Clark et al., 2024; Eyring et al., 2024) adopt a unified global objective by minimizing a surrogate loss of the form

$$\mathcal{L}(x) = C - \sum_{k=1}^{K} w_k R^k(x), \tag{1}$$

where $C$ is treated as a constant upper bound on the aggregated reward. However, adopting a global upper bound $C$ introduces a fundamental mismatch with real-world settings, where different prompts inherently admit different maximum achievable rewards, as illustrated in Fig. 1. Moreover, each reward dimension varies not only in scale but also in optimization difficulty across prompts, giving rise to a highly heterogeneous optimization landscape. Enforcing a single global reward bound therefore ignores prompt-specific feasible maxima, resulting in biased gradient signals that

encourage shortcut exploitation and ultimately lead to reward hacking. We formally show that under heterogeneous prompt-wise reward bounds, optimizing toward a unified global target can systematically induce reward hacking for a subset of samples, as detailed in Appendix C.

## 2.2. Reward Hacking via Weak Reward Models

In addition to heterogeneous reward bounds, the quality of individual reward models is another critical factor contributing to reward hacking. Using two high-quality, peer-reviewed human-preference datasets, Pick-a-Pic(Kirstain et al., 2023) and Pick-High(Ba et al., 2025), we evaluate the preference prediction capability of different reward models (Table 1). Based on their accuracy in predicting human preferences, we categorize reward models into *strong* and *weak* according to their degree of alignment with human-perceived image quality, where strong rewards exhibit consistently higher human-preference prediction accuracy.

Based on this distinction, we hypothesize that weak reward models are more prone to reward hacking, whereas strong reward models provide more stable and reliable optimization signals. Intuitively, weak reward models exhibit lower alignment with human perceptual judgment and are therefore more susceptible to shortcut exploitation. During optimization, models may amplify superficial or spuriously correlated features favored by weak rewards to rapidly increase reward scores, rather than improving overall perceptual quality. Such shortcut-driven optimization leads to reward hacking, where artificial score gains replace genuine quality improvements. In contrast, strong reward models more consistently reflect human preferences across prompts, offering more coherent and constraining feedback signals. This consistency is expected to suppress shortcut exploitation and guide multi-reward optimization toward meaningful and stable quality improvements. Formal definitions and correlation-based criteria are provided in Appendix C.

## 2.3. Quantitative Detection of Reward Hacking

Reward hacking is often diagnosed through human visual inspection, yet such qualitative assessments are inherently unreliable and difficult to quantify. Importantly, when reward hacking occurs in one reward dimension, the corresponding reward score may continue to increase, while other reward models can detect the abnormality and assign degraded scores. This cross-reward imbalance allow population-level averages to remain misleadingly positive, rendering standard mean-based evaluation insufficient for reliably identifying reward hacking. Motivated by this property, we propose two principled metrics, the Joint Domination Rate (JDR) and the Joint Collapse Rate (JCR), to explicitly reflect the presence of reward hacking and to evaluate multi-reward optimization behavior. **Joint Domination Rate (JDR).** JDR measures

*Table 1.* Human preference prediction accuracy (%) on high-quality image datasets,Pick-a-Pic(Kirstain et al., 2023) and Pick-High(Ba et al., 2025). **Bold** indicates *strong* reward models with higher human-preference prediction accuracy.

| Reward | CLIP | HPS | ICT | HP |
|---|---|---|---|---|
| Accuracy (%) | 60.30 | 72.88 | **87.58** | **88.47** |

the proportion of samples that simultaneously improve over a baseline across all $K$ rewards:

$$\text{JDR}_K = \frac{1}{N} \sum_{i=1}^{N} \mathbb{K}(\mathbf{R}_i \succ \mathbf{R}_{i,b}), \qquad (2)$$

where $\mathbf{R}_i \in \mathbb{R}^K$ and $\mathbf{R}_{i,b}$ denote the reward vectors of sample $i$ and its baseline, respectively.

**Joint Collapse Rate (JCR).** JCR measures the fraction of samples whose reward vectors are strictly dominated by the baseline across all $K$ rewards:

$$\text{JCR}_K = \frac{1}{N} \sum_{i=1}^{N} \mathbb{K}(\mathbf{R}_{i,b} \succ \mathbf{R}_i), \qquad (3)$$

where $\mathbf{R}_i \in \mathbb{R}^K$ and $\mathbf{R}_{i,b}$ denote the reward vectors of sample $i$ and its baseline, respectively.

JDR evaluates the effectiveness of multi-reward optimization while detecting reward hacking by enforcing consistent improvement across all reward dimensions. Shortcut-induced inflation in any single reward is counteracted by penalties from other rewards, preventing spurious gains from being reflected in JDR. In contrast, JCR reveals hidden collapse patterns that mean-based metrics fail to capture. Together, these metrics provide a robust and faithful diagnostic of reward hacking beyond population-level averages.

## 3. Pareto-Frontier-Guided Optimal Transport

### 3.1. Multi-reward Optimization Formulation

Multi-reward optimization aims to achieve synergistic improvements across multiple reward models. Since objectives may be conflicting, the problem naturally falls within the framework of Pareto optimization. In the Pareto framework, "conflict" primarily refers to unavoidable trade-offs near the Pareto frontier: while multiple rewards can improve together in suboptimal regions, further improving one reward close to optimality often requires sacrificing at least one other. As shown in Table 2, optimizing solely for CLIP model improves alignment-related metrics (CLIP and ICT) while simultaneously degrading human preference metrics (HPS and HP), highlighting the intrinsic conflicts among rewards. Motivated by these inherent trade-offs, we next introduce the fundamental concepts of Pareto optimality under multiple reward signals.

**Pareto Optimality Fundamentals.** Given $K$ rewards $\tilde{R} = (R^1(x), R^2(x), \ldots, R^K(x))$ to be maximized, the concepts of Pareto optimality can be defined as follows:

- **Pareto Dominance.** The reward vector of $x^a$ *dominates* that of $x^b$ (denoted $\tilde{R}(x^a) \succ \tilde{R}(x^b)$) if

$$\forall u \in \{1, \ldots, K\} : R^u(x^a) \geq R^u(x^b)$$
$$\wedge \; \exists v \in \{1, \ldots, K\} : R^v(x^a) > R^v(x^b).$$

- **Pareto Optimality.** For a given prompt $p_i$, a sample $x^* \in \mathcal{F}_i$ is *Pareto optimal* if its reward vector $\tilde{R}(x^*)$ is not dominated by that of any other $x \in \mathcal{F}_i$:

$$x^* \text{ is Pareto optimal}$$
$$\Leftrightarrow x^* \in \mathcal{F}_i \wedge \nexists x \in \mathcal{F}_i : \tilde{R}(x) \succ \tilde{R}(x^*).$$

- **Pareto Front.** Given prompt $p_i$, the *Pareto front* $\mathcal{J}_i$ is the set of all samples in the perceptually admissible feasible set $\mathcal{F}_i$ whose rewards are Pareto optimal:

$$\mathcal{J}_i = \{x \mid x \in \mathcal{F}_i, \; \nexists x' \in \mathcal{F}_i : \tilde{R}(x') \succ \tilde{R}(x)\}.$$

To tackle the issues beyond reward hacking identified in Section 2, we propose a Pareto-guided optimal transport framework for multi-reward optimization. First, an offline strategy leverages precomputed Pareto fronts for each prompt domain to mitigate reward hacking arising from heterogeneous bounds (Section 2.1). Second, addressing the assumption that weak rewards are prone to collapse (Assumption C.4), we introduce a GPT-4o-based decision agent to detect and eliminate unstable weak rewards. Third, an online strategy enables strong rewards to autonomously explore and expand the Pareto front during optimization. Finally, we propose JDR and JCR as metrics to evaluate performance gains and stability in multi-reward optimization.

### 3.2. Offline Pareto Frontier Guidance Strategy

To address the reward hacking issue caused by the global bound, as discussed in Section 2.1, we precompute prompt-specific candidate reward sets to estimate heterogeneous reward bounds before training. Specifically, for a given prompt $p_i$, the T2I model generates $M$ candidate samples $\{x_i^j\}_{j=1}^M$, forming the precomputed candidate set $\mathcal{R}_{i,M}^{(\text{pre})} = \{\tilde{R}(x_i^j) \mid j = 1, \ldots, M\}$. We then adopt dominance matrix–based non-dominated sorting to extract the Pareto frontier. For the $M$ candidate reward vectors in $\mathcal{R}_{i,M}^{(\text{pre})}$, we build an $M \times M$ dominance matrix $A$, with $A_{mn} = 1$ if $x_i^m$ dominates $x_i^n$ (i.e., $\tilde{R}(x_i^m) \succ \tilde{R}(x_i^n)$), and 0 otherwise. Reward vectors with a domination count of zero (that is, those not dominated by any others) constitute the Pareto frontier:

$$\mathcal{R}^{front}(p_i) = \{\tilde{R}(x_i^j) \in \mathcal{R}_{i,M}^{(\text{pre})} \mid \sum_{m=1}^M A_{mj} = 0\}, \quad (4)$$

*Table 2.* Text–image alignment and human preference score trends during CLIP-only optimization, revealing conflicts among reward signals. $\Delta_n$ reports the percentage change with respect to the original model at optimization step $n$.

| Metric | Base | $\Delta_{100}$ | $\Delta_{200}$ | $\Delta_{300}$ | $\Delta_{400}$ |
|--------|------|--------|--------|--------|--------|
| CLIP | 0.29 | +0.03% | +3.40% | +6.56% | +7.27% |
| ICT | 0.77 | +0.00% | +0.26% | +4.45% | +6.16% |
| HPS | 0.26 | -0.00% | -0.08% | -2.58% | -2.78% |
| HP | 0.78 | -0.00% | -0.05% | -1.00% | -4.38% |

where $q_i = |\mathcal{R}^{front}(p_i)|$ denotes the number of Pareto-optimal points on the Pareto frontier for prompt $p_i$.

During training, for each prompt $p_i$, we retrieve its Pareto frontier $\mathcal{R}^{front}(p_i)$ and generate samples with the T2I model to obtain reward vectors dominated by all points on the frontier. The Pareto frontier serves as the optimization target, guiding these dominated rewards toward Pareto-optimal solutions. This is formalized by minimizing the optimal transport discrepancy between the distributions of dominated reward vectors and Pareto frontier reward vectors.

**Optimal Transport Framework.** Optimal Transport (OT) (Monge, 1781) provides a principled way to measure discrepancies between probability distributions while preserving the geometry of the underlying space. Given a source distribution $\mu$ and a target distribution $\nu$, OT seeks a transport plan $\gamma \in \Pi(\mu, \nu)$ that minimizes the total cost:

$$\min_{\gamma \in \Pi(\mu,\nu)} \int c(x, y) \, d\gamma(x, y), \quad (5)$$

where $c(x, y)$ denotes the ground cost between source sample $x$ and target sample $y$. Here, the $n$ rewards dominated by the frontier collectively constitute the source distribution: $\mu_i = \{\tilde{R}(x_i^j) \mid x_i^j \in \{x_i^1, \ldots, x_i^n\}, \forall \tilde{R}(x_i^m) \in \mathcal{R}^{(q)}(p_i) : \tilde{R}(x_i^m) \succ \tilde{R}(x_i^j)\}$, while the precomputed Pareto frontier serves as the target distribution $\nu_i = \mathcal{R}^{(q)}(p_i)$.

OT establishes a minimal-cost mapping from $\mu_i$ to $\nu_i$, transporting dominated samples toward dominating points according to geometric distances in the reward space for prompt $p_i$. Practically, the optimal plan $\gamma_i^*$ is solved by optimizing the following discrete formulation of OT with an entropy regularization term using the Sinkhorn algorithm (Cuturi, 2013):

$$\gamma_i^* = \arg \min_{\gamma \in \Pi(\mu_i, \nu_i)} \sum_{m,j} c(y_i^m, x_i^j) \, \gamma(y_i^m, x_i^j), \quad (6)$$

where the ground cost is defined as $c(y_i^m, x_i^j) = \|\tilde{R}(y_i^m) - \tilde{R}(x_i^j)\|_2^2$, representing the squared Euclidean distance between reward vectors of source sample $y_i^m$ and target sample

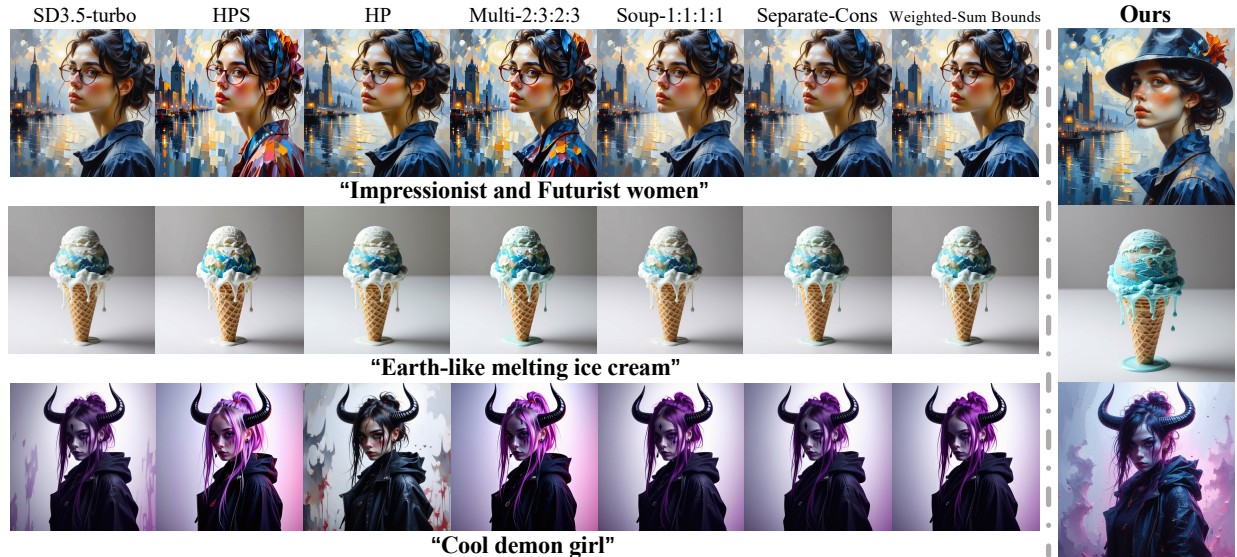

| SD3.5-turbo | HPS | HP | Multi-2:3:2:3 | Soup-1:1:1:1 | Separate-Cons | Weighted-Sum Bounds | **Ours** |

**"Impressionist and Futurist women"**

**"Earth-like melting ice cream"**

**"Cool demon girl"**

*Figure 2.* Qualitative Comparison of Optimization Results Across Different Methods.

$x_i^j$ whose reward vector is in the Pareto frontier, and the $\gamma$ is a $n \times q_i$ transport matrix.

**Training Objective.** In practice, $\tilde{R}(x)$ is computed by reward models on the generated image $x$. We optimize the T2I model parameters by backpropagating the OT-based loss through the reward computation, following the differentiable reward optimization setup (e.g., DRaFT-K) used in our training pipeline. This offline strategy is applicable to multi-reward training scenarios containing suboptimal reward models prone to shortcut behaviors, providing the model with precomputed Pareto frontiers as optimization targets to guide collaborative robust optimization across multiple reward models.

### 3.3. VLM-based Decision-making Agent

Our experiments confirm the hypothesis (Section 2.2) that weak reward models are susceptible to reward hacking. To mitigate this, we employ a **vision-language model (VLM)** as a decision-making agent for detecting and removing collapsed weak reward models. Once significant signals of reward hacking occur, the optimization trajectory becomes difficult to recover, making early detection critical. However, in the early stages of reward hacking, the generated images are only mildly collapsed, with only slight differences from normally generated images. To capture these subtle differences, we collected images of early mild collapses for each reward model and use a VLM to summarize collapse characteristics, constructing a prior reference for the agent. Upon detecting mild collapse, the agent immediately removes the problematic weak reward model and reverts to an earlier stable checkpoint to safely resume training.

### 3.4. Online Pareto Domination Strategy

After the decision-making agent eliminates all weak reward models prone to reward hacking, the remaining strong reward models exhibit a strong correlation between their preference prediction capabilities and human perceptual quality. To further explore superior reward bounds, we propose an online strategy that enables strong reward models to autonomously collect and optimize along the Pareto frontier during training. Specifically, given a prompt $p_i$, we perform the T2I model in parallel across multiple processes to increase the number of generated images. The reward vectors from all processes are aggregated into a global set $\mathcal{R}_{i,N} = \{\tilde{R}(x_i^j) \mid j = 1, \ldots, N\}$, where $n$ is the number of candidate images per process, $P$ is the number of processes, and $N = P \times n$.

For each sample $x_i^j$, we identify all vectors in $\mathcal{R}_{i,N}$ that Pareto-dominate $\tilde{R}(x_i^j)$:

$$\mathcal{R}^{dom}(x_i^j) = \left\{ \tilde{R}(x_i^m) \in \mathcal{R}_{i,N} \mid \tilde{R}(x_i^m) \succ \tilde{R}(x_i^j) \right\}, \quad (7)$$

where $k_i = |\mathcal{R}^{dom}(x_i^j)|$ is the number of dominating reward vectors. The reward vector of $x_i^j$ forms a discrete source distribution with one sampling point $\mu = \tilde{R}(x_i^j)$, while the dominating set serves as target distribution $\nu = \mathcal{R}^{dom}(x_i^j)$. For each batch, the optimal transport discrepancies between the source and target distributions for all samples are computed and summed to obtain the batch loss.

As training progresses and the reward candidate set improves, the online strategy adaptively guides strong reward models to explore superior Pareto frontiers, thereby achieving better optimization.

*Table 3.* Quantitative Results (%) on the Parti-Prompts dataset (Yu et al., 2022): Individual Reward Win Rates and Joint Performance Metrics Relative to SD3.5-Turbo.

| Model | ICT (%)↑ | HP (%)↑ | CLIP (%)↑ | HPS (%)↑ | JDR$_2$ (%)↑ | JDR$_4$ (%)↑ | JCR$_4$ (%)↓ |
|---|---|---|---|---|---|---|---|
| **Single-Reward Optimization** | | | | | | | |
| + ICT | **56.99** | 36.83 | 47.06 | 48.71 | 20.59 | 7.66 | 10.17 |
| + HP | 52.45 | **90.26** | 44.30 | 57.29 | 36.15 | 13.73 | 4.11 |
| + CLIP | 48.96 | 47.06 | **52.63** | 46.81 | 23.77 | 8.09 | 9.07 |
| + HPS | 50.12 | 41.67 | 37.07 | **88.30** | 20.77 | 8.03 | 3.06 |
| **Multi-Reward Joint Optimization** (Weighted **ICT:HP:CLIP:HPS** ratios) | | | | | | | |
| 1:1:1:1 | 51.10 | 52.08 | 47.43 | 82.97 | 26.59 | 12.68 | 3.19 |
| 2:3:2:3 | 50.80 | 56.43 | 46.51 | 86.03 | 28.31 | 13.42 | 2.57 |
| 2:2:3:3 | 50.43 | 56.56 | 46.57 | 84.25 | 26.23 | 12.62 | 2.76 |
| 4:4:1:1 | 51.96 | 57.23 | 44.12 | 79.90 | 29.53 | 12.44 | 3.74 |
| **Reward Soup** (Weighted **ICT:HP:CLIP:HPS** fusion of single-reward LoRAs) | | | | | | | |
| 1:1:1:1 | 50.55 | 54.17 | 42.16 | 81.92 | 27.02 | 11.15 | 3.74 |
| 1:1:4:4 | 50.43 | 52.94 | 42.46 | 85.11 | 25.37 | 11.15 | 3.31 |
| 3:2:1:4 | 50.80 | 53.74 | 43.32 | 85.29 | 26.29 | 10.85 | 3.19 |
| 3:2:0:0 | 50.74 | 53.86 | 42.59 | 83.21 | 26.10 | 10.85 | 3.31 |
| **Multi-Reward under Heterogeneous Bounds** (w/o OT) | | | | | | | |
| Weighted-Sum | 52.63 | 56.86 | 46.94 | 82.48 | 29.84 | 13.66 | 3.49 |
| Separate-Cons | 49.45 | 57.23 | 46.63 | 61.21 | 28.25 | 10.78 | 6.68 |
| **Pareto-Frontier-Guided Optimal Transport** | | | | | | | |
| PG-OT | 56.43 | 85.23 | 43.63 | 61.70 | **47.98** | **17.10** | **2.39** |

*Table 4.* Agreement accuracy between VLM-based reward hacking detection and human expert assessments across 200 evaluation samples. The "/T" indicates the thinking-mode variants.

| Model | GPT-4o | Qwen3-VL-32B/T | GLM-4.5V |
|---|---|---|---|
| Accuracy | 90.5% | 87.5% | 84.0% |

## 4. Experiments

### 4.1. Experiment Setting

**Implementation Details.** Our text-to-image (T2I) framework is built upon Stable Diffusion 3.5-Turbo, a state-of-the-art text-to-image model. To ensure training stability, we fine-tune only LoRA (Hu et al., 2021) parameters while keeping the original model weights frozen. Our method is compatible with a broad class of approaches that directly backpropagate gradients from reward models; in this work, we adopt the DRaFT-K strategy (Clark et al., 2024), which applies gradient updates only during the final $k = 2$ denoising steps to reduce memory consumption and accelerate training. The initial Pareto frontier is constructed by generating $M = 50$ images per prompt, providing a sufficiently tight approximation of prompt-specific reward upper bounds in practice. More training details are in Appendix G.

**VLM-based Decision Making Agent.** Our reward-hacking detection agent is implemented using vision-language models (VLMs) equipped with a carefully designed and robust chain-of-thought (CoT) reasoning process. To assess both accuracy and reproducibility, we evaluate one closed-source VLM (GPT-4o) and multiple open-source VLMs, including Qwen3-VL-32B-Thinking (Bai et al., 2025) and GLM-4.5V (Team et al., 2026) on 200 human-expert-annotated evaluation samples, as shown in Table 4. All evaluated VLMs exhibit strong agreement with human expert judgments, demonstrating that the proposed agent can be reliably instantiated using either closed- or open-source VLMs with consistent performance and reproducibility. For implementation convenience, we adopt GPT-4o as the agent during training. The agent is invoked only at fixed intervals, and each detection pass costs $0.015, accounting for merely 0.4% of the total training cost. A detailed description of the agent design and workflow is provided in Appendix D.

**Computational Complexity.** Our method introduces a one-time Pareto frontier precomputation prior to training, which takes 2 hours (about 12% of the total training time) and yields better-calibrated optimization targets, enabling 5× faster convergence. During training, the per-iteration runtime increases only marginally from 6.0 s for the baseline to 6.18 s for our method (3% overhead), where the OT-based loss computation contributes less than 0.2 s per iteration and Pareto frontier online extraction incurs only 0.4 ms, accounting for a negligible fraction of the total iteration time. Overall, the proposed framework incurs minimal additional computational cost while achieving substantially more stable training and effective reward hacking mitigation.

*Table 5.* Quantitative Results on the Parti-Prompts dataset: Comparison of multiple reward scores.

| Model | CLIP↑ | ICT↑ | Aesthetic↑ | HPS↑ | PickScore↑ | ImgReward↑ | HP↑ |
|---|---|---|---|---|---|---|---|
| SD3.5-Turbo | 0.3372 | 0.8965 | 6.2766 | 0.2856 | 22.7435 | 1.1499 | 0.7754 |
| **Single-Reward Optimization** | | | | | | | |
| ICT | 0.3548 | 0.8986 | 6.2781 | 0.2916 | 22.7261 | 1.1487 | 0.7739 |
| HP | 0.3538 | 0.8976 | 6.2881 | 0.2809 | 22.7687 | 1.1691 | 0.7776 |
| CLIP | **0.3554** | 0.8958 | 6.2770 | 0.2915 | 22.7396 | 1.1530 | 0.7751 |
| HPS | 0.3501 | 0.8987 | 6.3437 | **0.2994** | 22.6896 | 1.1793 | 0.7747 |
| **Multi-Reward Joint Optimization** (Weighted **ICT:HP:CLIP:HPS** ratios) | | | | | | | |
| 1:1:1:1 | 0.3545 | 0.8970 | 6.2839 | 0.2958 | 22.6895 | 1.1464 | 0.7763 |
| 2:3:2:3 | 0.3544 | 0.8987 | 6.2850 | 0.2971 | 22.6835 | 1.1636 | 0.7763 |
| 2:2:3:3 | 0.3552 | 0.8974 | 6.2787 | 0.2962 | 22.6926 | 1.1570 | 0.7770 |
| 4:4:1:1 | 0.3541 | 0.8989 | 6.3043 | 0.2950 | 22.7049 | 1.1568 | 0.7762 |
| **Reward Soup** (Weighted **ICT:HP:CLIP:HPS** fusion of single-reward LoRAs) | | | | | | | |
| 1:1:1:1 | 0.3543 | 0.8958 | 6.2931 | 0.2936 | 22.7542 | 1.1562 | 0.7752 |
| 1:1:4:4 | 0.3539 | 0.8967 | 6.3037 | 0.2951 | 22.7559 | 1.1679 | 0.7752 |
| 3:2:1:4 | 0.3537 | 0.8965 | 6.3032 | 0.2951 | 22.7549 | 1.1654 | 0.7754 |
| 3:2:0:0 | 0.3541 | 0.8961 | 6.2759 | 0.2941 | 22.7436 | 1.1493 | 0.7752 |
| **Multi-Reward under Heterogeneous Bounds** (w/o OT) | | | | | | | |
| Weighted-Sum | 0.3546 | 0.8968 | 6.2800 | 0.2947 | 22.7274 | 1.1514 | 0.7760 |
| Separate-Cons | 0.3549 | 0.8967 | 6.2774 | 0.2922 | 22.7343 | 1.1561 | 0.7757 |
| **Pareto-Frontier-Guided Optimal Transport** | | | | | | | |
| PG-OT | 0.3534 | **0.9004** | **6.3588** | 0.2929 | **22.8160** | **1.1808** | **0.7783** |

## 4.2. Baseline Construction

To fairly evaluate our multi-reward optimization framework, we construct four baselines: single-reward fine-tuning, weighted multi-reward fine-tuning, reward soup, and multi-reward fine-tuning under heterogeneous reward bounds. These baselines cover common training-time and inference-time strategies for multi-reward optimization. All methods use identical hyperparameters, and we report the best checkpoint for each baseline. Detailed baseline descriptions are in Appendix G. Below, we focus on the heterogeneous reward bounds baseline and describe its setup in detail.

**Multi-Reward Fine-Tuning with Heterogeneous Reward Bounds.** We utilize the precomputed Pareto frontier as an approximation of prompt-wise heterogeneous reward bounds, and build two baseline methods on top of it that rely solely on simple mapping, without employing optimal transport. One variant aggregates approximate bounds of multiple reward functions into a weighted average, serving as a unified optimization target across prompts, denoted as the *weighted-sum bounds* method. The other assigns each reward bound as its optimization target and minimizes the squared error between each reward and its bound, formulated as $\mathcal{L} = \sum_k (r_k^{\mathrm{bound}} - r_k)^2$, denoted as the *separate-constraints* method.

## 4.3. Evaluation and Analysis

**Joint Performance Metrics.**

In Table 3, we report the individual win rates of four reward models (ICT, HP, CLIP, and HPS), together with their joint optimization performance on the Parti-Prompts dataset (Yu et al., 2022). We evaluate joint optimization using three metrics: $JDR_2$, which measures the joint domination rate on the two strong rewards (ICT and HP) selected by the agent; $JDR_4$, which assesses the joint domination rate across all four rewards; and $JCR_4$, which quantifies the joint collapse rate over all rewards, where lower values indicate better stability. Our method demonstrates significant improvements over the baselines, achieving an 11% gain in $JDR_2$, a 3.4% gain in $JDR_4$, and a 0.2% reduction in $JCR_4$, while maintaining comparable win rates on each individual reward. Single-reward baselines achieve the highest win rates on their respective rewards, but their joint performance metrics generally degrade.

**Reward Metrics.**

Table 5 presents comprehensive evaluation results across seven reward models. Our method consistently secures the best performance on most metrics, demonstrating robustness under heterogeneous bounds and strong-reward training. In contrast, multi-reward joint optimization methods are affected by weak rewards, leading to degraded overall

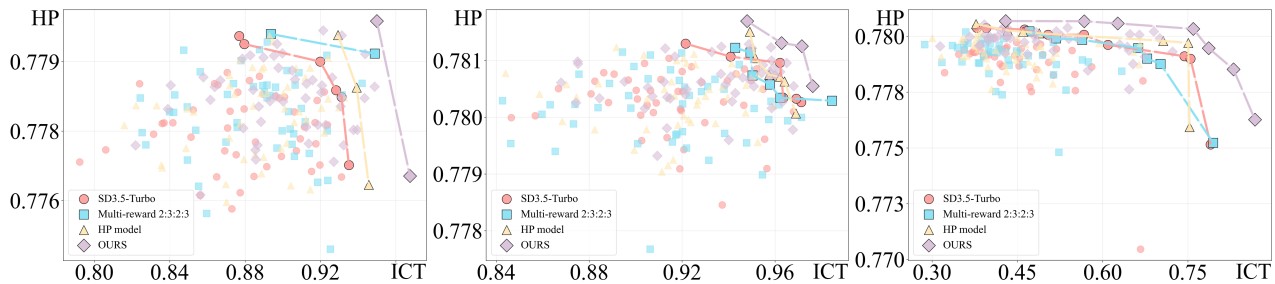

*Figure 3.* Pareto Frontier Visualization based on strong rewards (ICT and HP) on Three Prompts.

*Table 6.* Win Rate (%) of Ours vs. Baselines by Human Experts on DiffusionDB (Wang et al., 2023).

| Model | SD3.5-Turbo | HP | HPS | Soup-1:1:1:1 | Multi-2323 | Separate-Constraints | Weighted-Sum Bounds |
|---|---|---|---|---|---|---|---|
| **Ours** vs. | 76.34 | 61.29 | 74.19 | 74.19 | **79.57** | 68.82 | 73.12 |

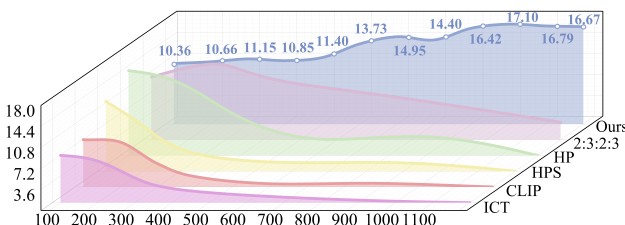

*Figure 4.* Comparative Training Curves of Joint Domination Rate (JDR$_4$) for Ours versus Baseline Methods.

*Table 7.* Ablation Study on JDR and JCR.

| Model | JDR$_2$↑ | JDR$_4$↑ | JCR$_4$↓ |
|---|---|---|---|
| SD3.5-Turbo | – | – | – |
| Online Only | 21.51 | 9.38 | 4.04 |
| Offline Only | 34.07 | 7.35 | 7.23 |
| Ours (w/o OT) | 18.15 | 10.78 | 6.68 |
| Ours (w/o Online) | 38.54 | 14.89 | 4.29 |
| **Ours** | **47.98** | **17.10** | **2.39** |

performance and instability.

**Quantitative results.**

As shown in Fig. 2, we present the qualitative comparisons. The baselines include two single-reward optimizations aligned with human preference, HPS and HP; a multi-reward joint training approach with the global bound as the optimization target using the weighted ratio ICT:HP:CLIP:HPS = 2:3:2:3; a reward-soup fusion method applied at inference with equal weighting ICT:HP:CLIP:HPS = 1:1:1:1; and two heterogeneous-bound methods, Weighted-Sum bounds and Separate-Constraints. The results demonstrate that our method achieves superior visual quality.

**Qualitative Case Studies on Pareto Frontier Visualization.** We conduct a Pareto frontier analysis based on two strong rewards, ICT and HP, comparing SD 3.5-Turbo, the HP-optimized single-reward model, the multi-reward joint optimization model with ratio ICT:HP:CLIP:HPS = 2:3:2:3, and our proposed method. For each prompt and identical random seed, every method generates 50 images, whose reward distributions and corresponding Pareto frontiers are plotted in a two-dimensional diagram. As shown in Fig. 3, the image domains induced by different prompts exhibit heterogeneous reward bounds. Our method consistently produces Pareto frontiers that dominate those of the

baselines, with generated samples distributed closer to the frontier, reflecting superior trade-off alignment and validating the effectiveness of our approach.

**Visualization and Analysis of Training Curve.**

In Figure 4, we present the training curves of the Joint Domination Rate (JDR$_4$), which serves as a robust metric of joint optimization, for our method and the baselines. Single-reward baselines optimized with the global bound collapse rapidly, while multi-reward optimization with the global bound and the involvement of weak rewards also exhibits severe deterioration. In contrast, our method achieves stable and consistent improvements throughout training while effectively avoiding reward hacking.

**User study.** We conduct a user study with ten annotators on 300 randomly selected prompts from the DiffusionDB dataset (Wang et al., 2023). For each prompt, image pairs (ours vs. baseline) were presented in random order, and annotators evaluated prompt fidelity and visual appeal. As shown in Table 6, our method achieves higher win rates against all baselines, confirming its effectiveness.

**Ablation Study** We conduct extensive ablation studies, including variants with only the online strategy, only the offline strategy, without optimal transport mapping, and with the offline strategy combined with the VLM-based agent,

as shown in Table 7. The results indicate that the offline-only strategy achieves more stable performance gains than the online-only strategy, suggesting that the precomputed Pareto frontier provides more reliable optimization targets. Variants that remove either the optimal transport mapping or the online strategy achieve suboptimal performance, demonstrating the effectiveness of both the optimal transport mapping and the VLM-based agent components. In addition, we provide ablation experiments on the entropy regularization strength $\varepsilon$ in Appendix E to validate the robustness of our hyperparameter selection.

## 5. Conclusion

In this work, we show that reward hacking arises from unified global targets under heterogeneous reward bounds, and from the inherent vulnerability of weak reward models. To address this, we propose a Pareto-frontier-guided optimal transport framework with online/offline strategies, and introduce JDR and JCR as principled evaluation metrics. Our approach achieves consistent gains over strong baselines.

## Impact Statement

This paper presents work whose goal is to advance the field of Machine Learning. There are many potential societal consequences of our work, none which we feel must be specifically highlighted here.

## Acknowledgments

This work was supported in part by the National Natural Science Foundation of China No. 62376277, the Public Computing Cloud of Renmin University of China and the Fund for Building World-Class Universities (Disciplines) of Renmin University of China.

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

# A. Full Training Algorithm

In this section, we provide the complete training pipeline of the proposed Pareto-Frontier-Guided Optimal Transport framework for multi-reward alignment. The algorithm consists of three key components: (1) offline Pareto frontier construction for estimating prompt-specific heterogeneous reward bounds, (2) VLM-based weak reward detection and removal, and (3) online Pareto-guided optimal transport optimization for strong reward models.

The offline stage first precomputes Pareto frontiers for each prompt to provide stable optimization targets under heterogeneous reward bounds. During training, the VLM-based decision-making agent periodically detects reward hacking behaviors and removes unstable weak reward models. After weak rewards are filtered out, the online Pareto domination strategy dynamically explores superior Pareto-optimal solutions using strong rewards.

Algorithm 1 summarizes the complete training procedure.

---

**Algorithm 1** Pareto-Frontier-Guided Optimal Transport

---

**Input:** T2I model $G_\theta$, prompts $\mathcal{P}$, reward models $\mathcal{R}$, VLM agent $\mathcal{A}$, offline sample number $M$.
**Output:** Optimized parameters $\theta$.

---

**1. Offline Pareto Frontier Construction**
For each prompt $p_i \in \mathcal{P}$:
    Generate $M$ images $\{x_i^j\}_{j=1}^M \sim G_\theta(p_i)$
    Compute reward vectors $\tilde{R}(x_i^j)$
    Extract Pareto frontier: $\mathcal{F}_i = \{\tilde{R}(x_i^j) \mid \nexists m : \tilde{R}(x_i^m) \succ \tilde{R}(x_i^j)\}$
**2. Pareto-Guided Multi-Reward Training**
Initialize mode $\leftarrow$ OFFLINE
For each training step $t$:
    Sample prompt batch $\mathcal{B}$ and generate images $x_i \sim G_\theta(p_i)$
    Compute reward vectors $\tilde{R}(x_i)$
    If mode $=$ OFFLINE:
        Retrieve the precomputed Pareto frontier $\mathcal{F}_i$ for prompt $p_i$
        Identify reward vectors dominated by the frontier
        Construct source distribution: $\mu_i = \{\tilde{R}(x_i)\}$ from generated samples
        Use Pareto frontier as target distribution: $\nu_i = \mathcal{F}_i$
        Compute Sinkhorn OT mapping from $\mu_i$ to $\nu_i$
    Else:
        Generate multiple candidate images across parallel processes
        Aggregate all reward vectors into global reward set: $\mathcal{G}_i = \{\tilde{R}(x_i^j)\}_{j=1}^N$
        For each sample, identify Pareto-dominating reward vectors: $\mathcal{D}(x_i^j) = \{\tilde{R}(x_i^m) \in \mathcal{G}_i \mid \tilde{R}(x_i^m) \succ \tilde{R}(x_i^j)\}$
        Use current reward vector as source distribution: $\mu_i = \{\tilde{R}(x_i^j)\}$
        Use dominating reward set as target distribution: $\nu_i = \mathcal{D}(x_i^j)$
        Compute Sinkhorn OT mapping from $\mu_i$ to $\nu_i$
    Compute OT loss: $\mathcal{L}_{OT} = \sum_i \text{OT}(\mu_i, \nu_i)$
    Update parameters: $\theta \leftarrow \theta - \eta \nabla_\theta \mathcal{L}_{OT}$
    Every $T_{\text{detect}}$ steps:
        Invoke VLM agent $\mathcal{A}$
        If reward hacking is detected for weak reward $R^k$:
            Remove $R^k$ from $\mathcal{R}$
            Restore previous stable checkpoint
        If all remaining rewards are strong and stable, set mode $\leftarrow$ ONLINE

---

# B. Related Work

**Single-Reward Optimization for Text-to-Image Diffusion Models**

Reward-based optimization has emerged as a crucial paradigm for improving text-to-image diffusion models, where reward models provide supervision signals to enhance generation quality. Current research focuses on two objectives: **(1) Text–Image Alignment.** CLIP (Radford et al., 2021) pioneered cross-modal embeddings that capture vision–language semantics, and BLIP (Li et al., 2022) extended this paradigm with bidirectional mechanisms for stronger alignment evaluation. However, ICT (Ba et al., 2025) demonstrated that both models systematically undervalue high-quality images, motivating the development of refined metrics for more faithful text representation. **(2) Human Preference Alignment.** Recent efforts

have shifted toward modeling human perceptual preferences as rewards to better align generation with subjective judgments. Reward models such as ImageReward (Xu et al., 2023), HPS (Wu et al., 2023), PickScore (Kirstain et al., 2023), and HP (Ba et al., 2025) are trained on large-scale preference datasets to approximate human judgments. However, these models may yield conflicting signals. A central challenge, therefore, lies in how to effectively integrate and jointly optimize multiple rewards to reconcile such discrepancies.

**Multi-Reward Optimization and Pareto-Front Methods for Text-to-Image Diffusion**

To simultaneously improve text–image consistency and human preference alignment, previous studies have introduced multiple reward signals into text-to-image diffusion models. Basic approaches typically rely on linear weighting; for example, ReNO (Eyring et al., 2024) applies weighted fusion of multiple rewards solely to optimize the noise initialization stage, while TextCraftor (Li et al., 2024a) confines the weighted optimization to the text encoder. However, such simple weighting schemes are limited when handling conflicting rewards. To address this, several works have introduced the concept of the Pareto-Optimal: Parrot (Lee et al., 2024) refines prompts via a Prompt Expansion Network and selects sample pairs that dominate across all rewards for reinforcement learning; within diffusion-DPO (Wallace et al., 2024) frameworks, CaPO (Lee et al., 2025) calibrates different rewards to enhance training stability, and BalancedPO (Tamboli et al., 2025) aggregates diverse reward signals through majority voting. Nevertheless, these methods often require additional auxiliary modules or extensive relabeling of paired samples, and they have not thoroughly examined the underlying causes of reward hacking—the most critical issue in reward optimization. In contrast, our method eliminates the need for paired datasets, directly integrates multiple rewards, and employs optimal transport to guide batch samples toward the Pareto frontier, thereby achieving more balanced and robust alignment under conflicting objectives.

## C. Analysis of Global Upper Bounds and Weak Rewards in Multi-Reward Optimization

**Definition C.1** (Reward Hacking). Given prompt $p_i$ with its corresponding image domain $\mathcal{D}_i$ and $K$ reward functions $(R^1, R^2, \ldots, R^K)$, let $\overline{R_i^k} = \max_{x \in \mathcal{F}_i} R^k(x)$ be the $k$-th reward upper bound over the perceptually admissible feasible set $\mathcal{F}_i$, and let $\underline{Q_i} = \min_{x \in \mathcal{F}_i} Q(x)$ be the minimum human-perceived quality within $\mathcal{F}_i$. We define the set of *reward-hacked* images as:

$$\mathcal{H}_i := \left\{ x \in \mathcal{D}_i \,\middle|\, \exists k : R^k(x) > \overline{R_i^k} \;\wedge\; Q(x) < \underline{Q_i} \right\}. \tag{8}$$

That is, samples with **at least one** reward dimension exceeding its feasible reward upper bound, yet with human-perceived quality below the feasible minimum.

**Assumption C.2** (Heterogeneous Reward Upper Bounds within Admissible Sets). Each prompt $p_i$ induces a perceptually admissible domain $\mathcal{F}_i \subseteq \mathcal{D}_i$ associated with a distinct upper bound on the reward function. This can be formalized as the Heterogeneous Reward Upper Bounds property:

$$\exists\, p_i \neq p_j \;\text{ s.t. }\; \overline{R_i} := \sup_{x \in \mathcal{F}_i} R(x) \;\neq\; \overline{R_j} := \sup_{x \in \mathcal{F}_j} R(x), \tag{9}$$

where $\overline{R_i}$ denotes the upper-bound within $\mathcal{F}_i$.

This heterogeneity arises from prompt-dependent semantic constraints, biases in the reward model, and related factors. Consequently, no single global reward bound applies uniformly across all prompt-induced domains. Empirical visualizations further support this property, revealing distinct reward bounds across different prompts.

**Proposition C.3** (Global Bound Induces Reward Hacking). *For any image domain $\mathcal{D}_i$ induced by a prompt $p_i$, let $\overline{S_i} = \max_{x \in \mathcal{F}_i} \sum_{k=1}^{K} w_k R^k(x)$ denote the maximal weighted reward attainable within its feasible set $\mathcal{F}_i$. By Property C.2, domains exhibit heterogeneous reward upper bounds. Since the global constant $C$ is derived from a weighted combination of global reward upper bounds across all domains, there exist domains for which $C > \overline{S_i}$. When optimizing $\mathcal{L}(x) = C - \sum_{k=1}^{K} w_k R^k(x)$, the objective drives $\sum_{k=1}^{K} w_k R^k(x)$ beyond $\overline{S_i}$, typically by disproportionately increasing rewards in some dimensions. This results in $R^k(x) > R_i^{k*}$ for some dimension $k$, while simultaneously $Q(x) < \underline{Q_i}$ occurs. By Definition C.1, this implies $x \in \mathcal{H}_i$. Additional proofs are given in Appendix C.*

**Assumption C.4** (Weak Reward Models Are Prone to Reward Hacking). We hypothesize that weak reward models $R^W$ are prone to reward hacking, while strong reward models $R^S$ stabilize training.

Weak reward models $R^W$ poorly correlate with human perceptual judgment, creating exploitable shortcuts. Rather than improving genuine perceptual quality which is a complex and multifaceted challenge, text-to-image models would maximize

scores by targeting superficial features that these weak models overvalue. This leads to reward hacking, models chase artificial score improvements instead of authentic quality gains, ultimately degrading the generated images.

In contrast, strong reward models $R^S$ maintain consistent alignment with human preference, providing reliable guidance for stable training and ensuring the example towards meaningful quality improvements.

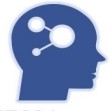

**VLM Agent**

You are a Strategy Decision Agent. I will provide you with two reference images:
the first image is a base image generated by sd3.5-turbo; the second image is a composite of four images optimized by different reward models, in the order from left to right: CLIP, ICT, HPS, and HP. These images exhibit reward hacking.

**Current environment variables:**
Reward models currently being trained: CLIP, HP, HPS, ICT
Training steps: 200
Reward models removed: None
Current training strategy: Offline strategy

sd3.5-turbo     CLIP    HP    HPS    ICT

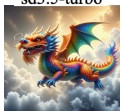 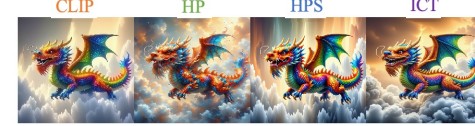

The characteristics of the breakdown images are as follows:
**CLIP** causes a breakdown with texture pixelation and detail degradation, making the image appear blurry.
**HP** results in edge artifacts and over-saturated color transitions.
**HPS** leads to over-saturation, sharpness distortion, and vertical stripe-like distortions.
**ICT** causes the lightest breakdown, characterized by smoother color transitions but a lack of depth.

*From VLM Thinking…*

**Your task is to make the next training decision based on the base image, the four composite breakdown images, and their corresponding descriptions, considering the current training environment variables:**
**1. Breakdown Mode Detection:**
  a）If any slight breakdown is observed, remove the reward model that caused the most obvious breakdown from {CLIP, ICT, HPS, HP} and revert to previous timestep.
  b）If no breakdown is observed, proceed to the next step of training strategy decision-making.
**2. Training Strategy Decision:**
Based on the current environment variables, decide whether to continue with the offline strategy or switch to the more flexible online strategy:
  a）If the remaining reward models can support stable training and generate images without breakdown, you may consider switching to the online strategy.
  b）Otherwise, continue with the offline strategy.

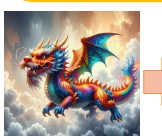

**Final Output**：The current image exhibits characteristics suggesting over-saturation and possible sharpness distortion similar to the effects produced by the **HPS model**. Thus, the HPS model likely causes reward hacking in this instance.
**Action**：**Remove the HPS reward model** from training and revert to the previous timestep. Maintain the **offline strategy for continued training**.

*Figure 5.* Adaptive Decision Pipeline of the VLM based Agent for Multi-Reward Optimization.

## D. Details VLM Based Decision-making Agent

We introduce VLM as a decision-making agent to adaptively manage multi-reward model training. The agent dynamically determines actions based on generated image quality and training stability, with three core capabilities:

- **Continue Training** — When no signs of collapse are observed in the generated images, the agent performs no additional operations and allows training to continue.

- **Remove & Revert** — Upon detecting a breakdown, the agent removes the unstable reward model and reverts to the most recent stable checkpoint.

- **Switch Strategy** — When training has proceeded for an extended period with minimal improvement in image optimization, while the remaining reward models remain stable, the agent smoothly transitions the process from offline training to online training.

### D.1. Prior Knowledge and Initialization

**Knowledge Base Construction.** For each reward model, we collect degraded image samples generated under breakdown conditions and employ VLM for semantic analysis to extract characteristic failure patterns. These include:

- **CLIP** — Texture pixelation and detail degradation, resulting in blurred appearance.

*Table 8.* Reward statistics exhibit heterogeneity across different prompts at training step 300.

*(a)* Mean reward.

| Reward | p1 | p2 | p3 |
|--------|--------|--------|--------|
| ICT | 0.8676 | 0.8114 | 0.9095 |
| CLIP | 0.2205 | 0.1421 | 0.2956 |
| HPS | 0.2449 | 0.2661 | 0.2749 |
| HP | 0.7800 | 0.7805 | 0.7805 |

*(b)* Standard deviation.

| Reward | p1 | p2 | p3 |
|--------|--------|--------|--------|
| ICT | 0.09 | 0.09 | 0.06 |
| CLIP | 0.03 | 0.04 | 0.03 |
| HPS | 0.007 | 0.01 | 0.007 |
| HP | 0.0006 | 0.0009 | 0.0005 |

*(c)* KL divergence.

| Reward | p1 | p2 | p3 |
|--------|--------|--------|--------|
| ICT | 8.00 | 5.45 | 11.42 |
| CLIP | 13.18 | 19.48 | 4.89 |
| HPS | 13.09 | 10.05 | 14.34 |
| HP | 11.04 | 10.78 | 16.82 |

- **ICT** — Mildest breakdown, characterized by overly smooth transitions and loss of depth.

- **HPS** — Severe over-saturation, sharpness distortion, and vertical stripe artifacts.

- **HP** — Pronounced edge artifacts and over-saturated color transitions.

The extracted patterns are structured into a prior knowledge base, which supports subsequent chain-of-thought (CoT)-based decision-making.

**State Initialization.** At the beginning of training, the agent is initialized with the following state information: (i) current training strategy (offline/online), (ii) active reward model set $\{R^i\}$, (iii) cumulative training step count $n$, and (iv) historical stability metrics with corresponding checkpoint references.

### D.2. Decision-Making Workflow

**Environment Context.** At each decision step, the agent receives structured contextual information, including: (i) the baseline image generated by the underlying model; (ii) composite breakdown images corresponding to different reward models (CLIP, ICT, HPS, HP); (iii) environment variables such as the set of active reward models, training step count, removal history, and the current strategy state (offline/online).

**Decision Procedure.** The agent performs a two-stage reasoning process:

1. **Breakdown Detection.** Compare diagnostic images with prior templates.

    - If a breakdown is detected, remove the most severely affected reward model and revert to the previous checkpoint.
    - If no breakdown is detected, proceed to the strategy evaluation stage.

2. **Training Strategy Decision.** Based on the current environment variables:

    - If the remaining reward models remain stable without breakdown, switch from the offline strategy to the more flexible online strategy.
    - Otherwise, maintain the offline training strategy.

**Final Output.** The decision agent produces a standardized output, which includes:

```
Final Output: [Training state analysis]
Action: [Remove / Revert / Continue]
Strategy Maintenance: [Offline / Online]
```

### D.3. Detailed Training Procedure

We implemented a staged dynamic optimization approach with systematic evaluation checkpoints every 100 steps to ensure stability in multi-reward training. The procedure consisted of two main phases: an initial offline phase with joint reward model training, during which weaker reward models were progressively removed until only stable ones remained, followed by the activation of the online strategy once convergence was achieved.

*Table 9.* Reward statistics for the same prompt across training steps. Human experts identified HPS-induced reward hacking beginning at step 200. $\Delta_n$ reports the percentage change with respect to the original model at optimization step $n$.

*(a)* Mean reward.

| Reward | $\Delta_{100}$ | $\Delta_{200}$ | $\Delta_{300}$ | $\Delta_{400}$ | $\Delta_{500}$ |
|---|---|---|---|---|---|
| ICT | 0.7835 | 0.8329 | 0.8729 | 0.8580 | 0.8658 |
| CLIP | 0.2879 | 0.2673 | 0.2200 | 0.1917 | 0.1592 |
| HPS | 0.2556 | 0.2528 | 0.2448 | 0.2296 | 0.2238 |
| HP | 0.7792 | 0.7796 | 0.7802 | 0.7799 | 0.7804 |

*(b)* Standard deviation.

| Reward | $\Delta_{100}$ | $\Delta_{200}$ | $\Delta_{300}$ | $\Delta_{500}$ | $\Delta_{500}$ |
|---|---|---|---|---|---|
| ICT | 0.1241 | 0.1180 | 0.0815 | 0.0857 | 0.0859 |
| CLIP | 0.0426 | 0.0416 | 0.0316 | 0.0245 | 0.0229 |
| HPS | 0.0079 | 0.0085 | 0.0070 | 0.0052 | 0.0064 |
| HP | 0.0011 | 0.0012 | 0.0007 | 0.0009 | 0.0007 |

*(c)* KL divergence.

| Reward | $\Delta_{100}$ | $\Delta_{200}$ | $\Delta_{300}$ | $\Delta_{400}$ | $\Delta_{500}$ |
|---|---|---|---|---|---|
| ICT | 5.9638 | **13.8767** | 5.9029 | 8.0059 | 9.6320 |
| CLIP | 5.5351 | **19.4165** | 4.9825 | 13.1837 | 18.0620 |
| HPS | 9.1868 | **19.9770** | 5.3309 | 13.0917 | 19.5950 |
| HP | 4.8112 | **14.1934** | 6.2217 | 11.0483 | 11.3300 |

**Offline Training with Adaptive Reward Management**   In the offline phase, four reward models (CLIP, ICT, HPS, HP) were trained simultaneously. At each 100-step interval, the VLM decision agent performed automated diagnostic analysis on composite output images to detect potential training instabilities. When anomalies were detected, the system reverted to the most recent stable checkpoint and adjusted the active reward set before resuming training.

The first breakdown occurred at step 200, where evaluation revealed over-saturation artifacts, sharpness distortions, and vertical stripe patterns consistent with HPS model instability. The agent removed HPS from the active reward set and rolled training back to step 100, after which training continued with the reduced set (CLIP, ICT, HP).

Training proceeded stably through steps 300, 400, and 500 without anomalies. However, at step 600, diagnostic analysis identified characteristic texture pixelation and detail degradation in CLIP outputs, while ICT and HP remained stable. The agent isolated CLIP as the instability source, removed it from the active set, and reverted to the step 500 checkpoint. Training then continued with the remaining reward pair (ICT, HP).

**Transition to Online Training**   Between steps 500 and 800, training exhibited sustained stability with no further breakdowns. After confirming at step 800 that generated images showed no collapse and only minimal changes, the system initiated the transition to online training mode for further exploration.

### D.4. Limitations of Heuristic Statistical Methods

Evaluation of Heuristic Methods for Detecting Reward Hacking. We conducted comprehensive experiments to evaluate whether heuristic methods could effectively detect reward hacking. Specifically, we analyzed three common statistical indicators: mean reward, reward variance (standard deviation), and KL divergence from a reference distribution. We evaluated these metrics under two scenarios: (1) images generated from different prompts at the same training step (Table 8), and (2) images generated from the same prompt across different training steps (Table 9).

Our results demonstrate that **no reliable heuristic pattern emerges** from these statistics:

**(1) Cross-prompt heterogeneity (Table 8):** We generated 50 images per prompt at step 300 (where reward hacking has occurred) and found that different prompts exhibit drastically different reward statistics. Since each training step samples

*Table 10.* Ablation study on entropy regularization strength $\varepsilon$ in Optimal Transport.

| Strength | ICT (%)↑ | HP (%)↑ | CLIP (%)↑ | HPS (%)↑ | JDR$_2$ (%)↑ | JDR$_4$ (%)↑ | JCR$_4$ (%)↓ |
|---|---|---|---|---|---|---|---|
| $\varepsilon = 0.1$ | 56.43 | 85.23 | 43.63 | 61.70 | 47.98 | 17.10 | 2.39 |
| $\varepsilon = 0.5$ | 57.50 | 80.41 | 51.70 | 69.00 | 49.20 | 16.50 | 3.50 |
| $\varepsilon = 0.9$ | 56.00 | 78.50 | 48.70 | 68.00 | 46.05 | 16.90 | 2.44 |

*Table 11.* Quantitative Results (%) on Parti-Prompts Dataset using ReFL.

| Model | ICT (%)↑ | HP (%)↑ | CLIP (%)↑ | HPS (%)↑ | JDR$_2$ (%)↑ | JDR$_4$ (%)↑ | JCR$_4$ (%)↓ |
|---|---|---|---|---|---|---|---|
| Baseline 2:3:2:3 | 51.05 | 51.50 | **50.50** | 64.50 | 24.50 | 9.24 | 7.45 |
| **Ours** | **52.08** | **61.46** | 49.69 | **68.32** | **31.86** | **13.54** | **4.72** |

different prompts, these statistical measures are fundamentally unstable and cannot serve as a basis for effective heuristic detection.

**(2) Reward indistinguishability (Table 9):** We measured reward statistics on images generated from the same prompt across different training steps (50 images per prompt), with human expert evaluation showing reward hacking caused by HPS beginning at step 200. The results reveal critical limitations of heuristic methods:

**Mean and standard deviation fail to detect anomalies (Tables 9(a) and (b)):** Neither metric exhibits clear abnormal patterns at the collapse point, making them unreliable for detecting.

**KL divergence detects collapse but cannot disentangle rewards (Table 9(c)):** While KL divergence shows sharp spikes that could signal reward hacking, all rewards exhibit similar sudden changes simultaneously. This makes it impossible to identify which specific reward is responsible for the collapse.

In contrast, our knowledge-driven agent successfully addresses both challenges: detecting when reward hacking occurs and identifying which reward causes it. These comprehensive experiments confirm that simple heuristic methods based on reward statistics are insufficient for detecting reward hacking in text-to-image generation, thereby motivating and validating our agent-based approach.

## E. Additional Quantitative Supplementary Experiments

### Ablation Study on Entropy Regularization Strength

We present an ablation study on the entropy regularization strength $\varepsilon$ in Table 10, demonstrating the robustness of our method across different hyperparameter settings.

**Generalizability Across Post-Training Methods.** Our method is broadly applicable to any gradient-based post-training approach. We designed our framework to be training-algorithm-agnostic, requiring only that the base method uses rewards to provide optimization signals. To demonstrate this generalizability, in addition to DRAFT-K presented in the main paper, we conduct additional experiments on ReFL (Xu et al., 2023)in Table 11.

**Scalability to Multiple Reward Models.** In terms of computational efficiency, weighted sum methods face a combinatorial explosion problem: When optimizing $n$ rewards using weighted sum methods, if each weight is selected from $m$ candidate values, the computational cost of weight selection grows combinatorially, with grid search complexity of $O(m^n)$, rendering it computationally infeasible when $n$ reaches dozens or hundreds. In contrast, our method leverages the distinguishability of collapse signatures across different reward models, achieving $O(n)$ complexity for reward hacking detection that scales linearly with the number of rewards. This linear scaling property makes our approach particularly advantageous for scenarios involving a large number of reward models, where traditional weighted sum approaches become prohibitively expensive.

## F. Additional Visualization Results

**Visualization and Analysis of Training Curve.** As shown in Figure 9, we present the joint domination rate JDR$_2$ on two strong rewards, ICT and HP. It can be observed that our method steadily improves as training progresses, whereas the baseline methods—whether single-reward optimization or multi-reward joint optimization—experience a rapid decline in

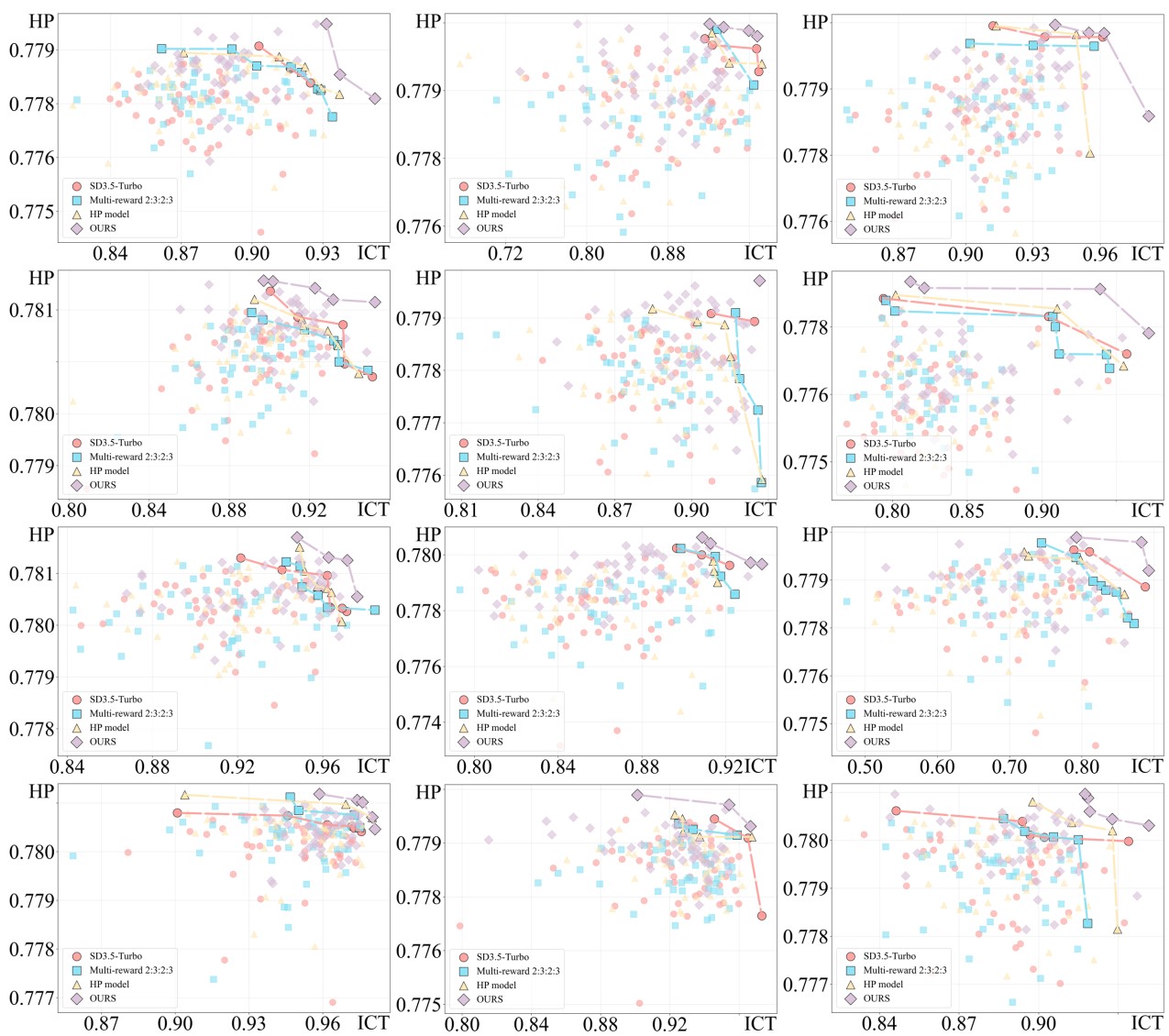

*Figure 6.* Broad Comparative Examples of Pareto Frontier Visualizations for Various Methods.

joint domination rate, indicating that the baseline approaches quickly encounter reward hacking issues.

**Qualitative Case Studies on Pareto Frontier Visualization.** As shown in Figure 6, we provide additional visualization examples of Pareto frontiers. It can be observed that the Pareto frontier characteristics vary across different prompts. Our method consistently dominates the baseline approaches, and the overall image distributions are closer to the Pareto frontier, demonstrating the superiority of our method.

**Visualization of Heterogeneous Reward Bounds Across Different Prompts** As shown in 7, we provide box plot visualizations of reward ranges for the ICT score across different prompts, based on 50 images generated with different seeds. In 8, we present the corresponding box plot visualizations for the HP score across different prompts. It can be observed from these extensive examples that the reward upper bounds are heterogeneous across prompts, and that the reward ranges and characteristics also vary between prompts.

**Visualization of Reward-Model-Specific Collapse Patterns.** Figure 10 presents representative collapse examples produced by four different reward models under multiple baseline optimization methods, with reward hacking types annotated by human experts. We observe that different reward models exhibit markedly distinct visual degradation patterns upon collapse. In contrast, for a given reward model, collapses induced under different optimization baselines consistently share

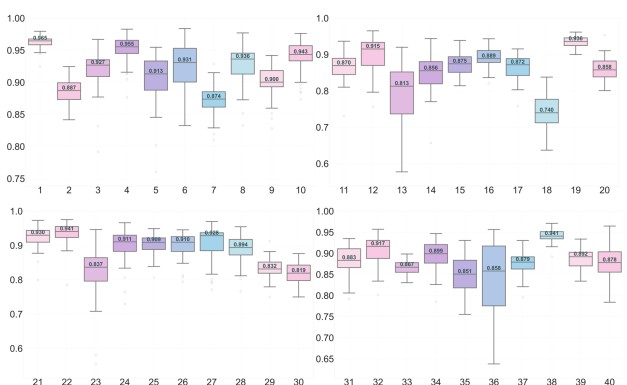

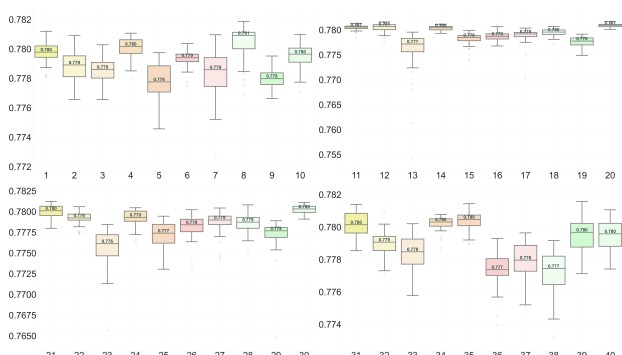

*Figure 7.* Visualization of Box Plots Showing Reward Variations Across Prompts on ICT Score.

*Figure 8.* Visualization of Box Plots Showing Reward Variations Across Prompts on HP Score.

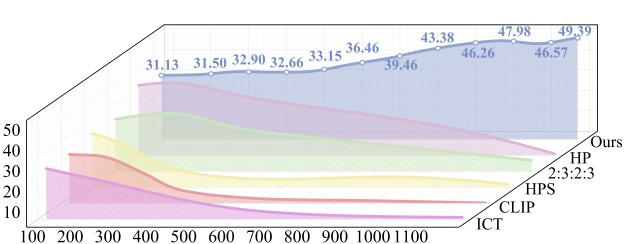

*Figure 9.* Training Curves of Joint Domination Rate ($JDR_2$).

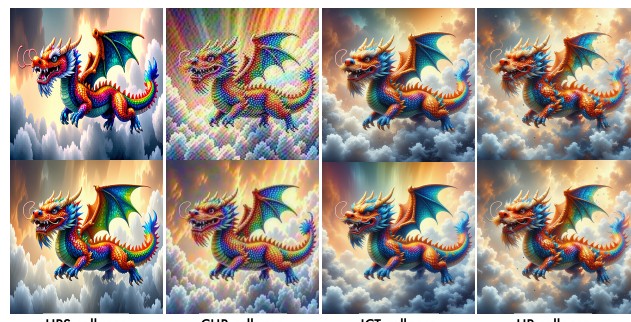

*Figure 10.* Visualization of Reward Hacking Across Diverse Reward Models.

highly similar structural characteristics. This observation indicates that reward hacking is not random noise, but rather a systematic failure mode driven by reward-model-specific preferences with identifiable visual patterns. Owing to the stability and reproducibility of these collapse signatures, the VLM-based decision agent is able to learn and recognize reward-model-specific collapse features, enabling accurate early-stage detection of reward hacking.

**More Qualitative Comparison Results.** We present additional visualizations for comprehensive comparison. Figure 11 shows results against single-reward baselines, while Figure 12 illustrates comparisons with multi-reward baselines.

## G. Experiment Details

For diffusion model optimization, we use 5,000 non-repetitive text prompts from the Pick-High dataset, a subset of the Pick-a-pic dataset. All experiments are conducted on a cluster of three nodes, each with eight A800 GPUs. We adopt DDIM sampling with four denoising steps, set the classifier-free guidance weight to $0.0$, and fix the output resolution to $512 \times 512$. The Adam optimizer is used for training with a learning rate of $5 \times 10^{-5}$.

### G.1. Baseline Construction

**Single-Reward Fine-Tuning.** Using Stable Diffusion 3.5-Turbo as the backbone, we fine-tune the model with CLIP, ICT, HPS, and HP as individual optimization objectives, employing DRaFT-K (Clark et al., 2024).

**Weighted Multi-Reward Fine-Tuning.** Using the weighted loss in Equation 1, we jointly optimize the four rewards(CLIP, ICT, HPS, and HP) under identical settings, systematically exploring diverse weight combinations.

**Reward Soup.** We adopt an inference-time fusion strategy, where the LoRA weights obtained from individual single-reward fine-tuned models are combined through weighted fusion. Specifically, this method dynamically merges parameters from multiple reward-specialized models during inference, thus exploring a broader reward fusion space without incurring

additional training costs.

### G.2. Computational Complexity

**Per-Iteration Computational Overhead Analysis.** Under the same experimental settings, the total runtime per training iteration increases marginally from $6.0\,\text{s}$ for the weighted-sum baseline to $6.18\,\text{s}$ for our method, corresponding to an overhead of approximately 3%. Among this overhead, the OT-based loss computation takes $161.17\,\text{ms}$ per iteration, compared to $81.34\,\text{ms}$ for the baseline loss. The additional $\sim\!80\,\text{ms}$ overhead is negligible in practice, as the backward pass of the diffusion model dominates the overall iteration time. A fine-grained breakdown further shows that Pareto frontier extraction incurs only $0.4\,\text{ms}$ per iteration, while optimal transport solving takes approximately $1.8\,\text{ms}$, both accounting for only a small fraction of the total iteration time.

**Computational Overhead of VLM-based Reward Hacking Detection.** The VLM-based reward hacking detection module is invoked once every 100 training steps. Each invocation takes approximately $12\,\text{s}$, which amortizes to only $0.12\,\text{s}$ per training iteration. This amortized overhead is small relative to the overall iteration time and does not significantly affect training efficiency in practice.

**Reward Models and Training Strategy.** We employ four reward models, encompassing both strong and weak categories, to initialize joint training. These are grouped into two primary types: text–image alignment rewards (**CLIP** (Radford et al., 2021) and **ICT** (Ba et al., 2025) ) and human preference rewards (**HPS** (Wu et al., 2023) and **HP** (Ba et al., 2025)). Our staged optimization strategy proceeds in three phases. First, all four reward models are jointly optimized during the offline policy stage. Next, weak rewards that induce instability or collapse are adaptively pruned according to agent feedback. Finally, the remaining strong reward models (**ICT** and **HP**) are retained to guide the online policy stage for deeper optimization.

**Evaluation Metrics.** To comprehensively assess the effectiveness of multi-reward optimization, we adopt Joint Performance metrics as the primary evaluation criterion, including the Joint Domination Rate (**JDR**) and Joint Collapse Rate (**JCR**). Specifically, we report $\text{JDR}_2$, computed on the two strong rewards ICT and HP that are ultimately retained by the agent's decision-making process, as well as $\text{JDR}_4$ and $\text{JCR}_4$, both computed over all four rewards ICT, HP, CLIP, and HPS to assess the overall optimization capability. In addition, we provide the average scores from seven widely used reward models as supplementary evaluation to validate the robustness of our approach: Aesthetic Score[1] for aesthetic quality, CLIP (Radford et al., 2021) for text–image consistency, ICT (Ba et al., 2025) for the degree of text presence in images, and human preference models such as PickScore (Kirstain et al., 2023), HPS (Wu et al., 2023), ImageReward (Xu et al., 2023), and HP (Ba et al., 2025).

---

[1]`https://github.com/christophschuhmann/improved-aesthetic-predictor`

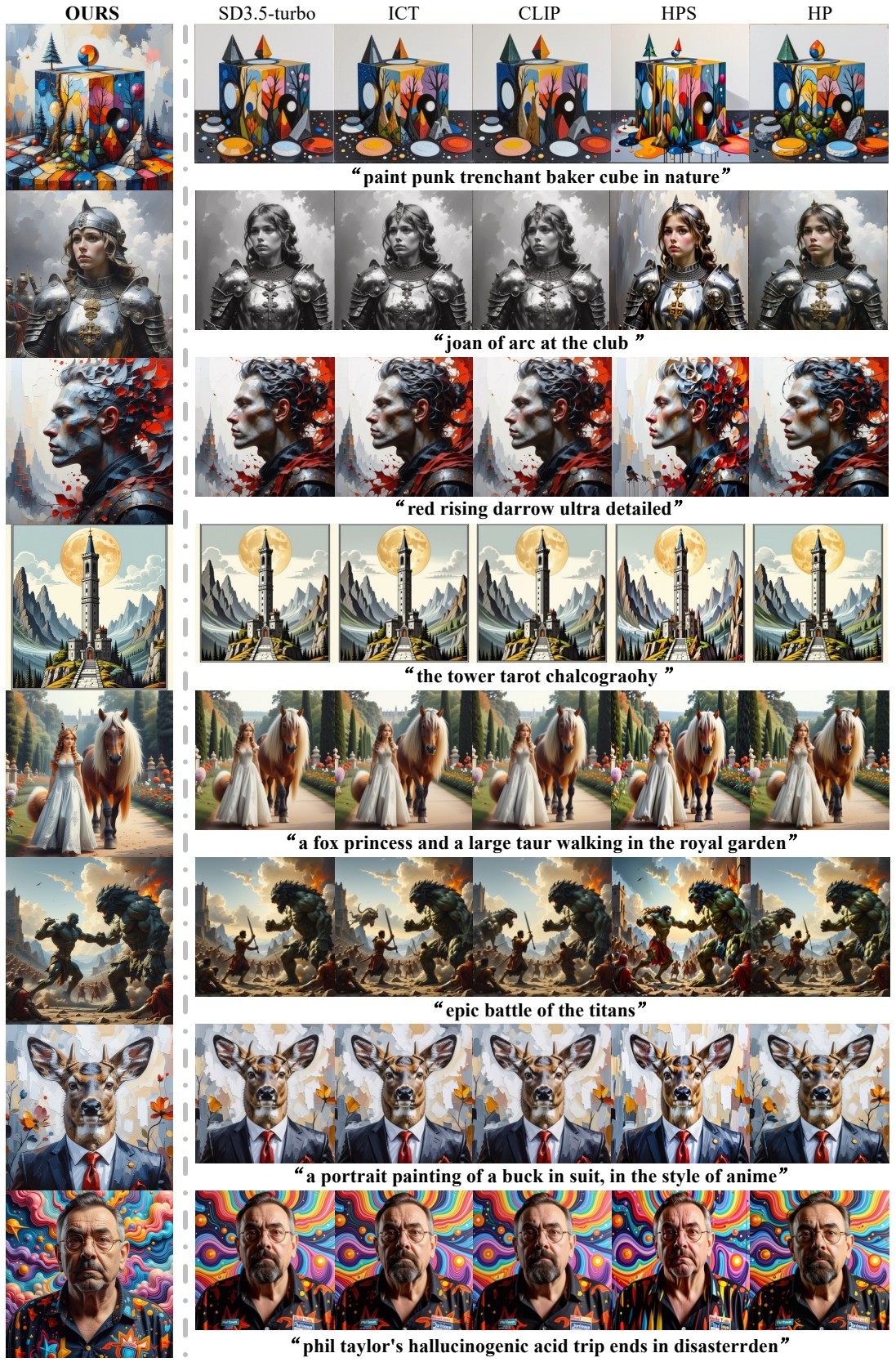

*Figure 11.* Qualitative Comparison of Optimization Results with Single-Reward Baselines.

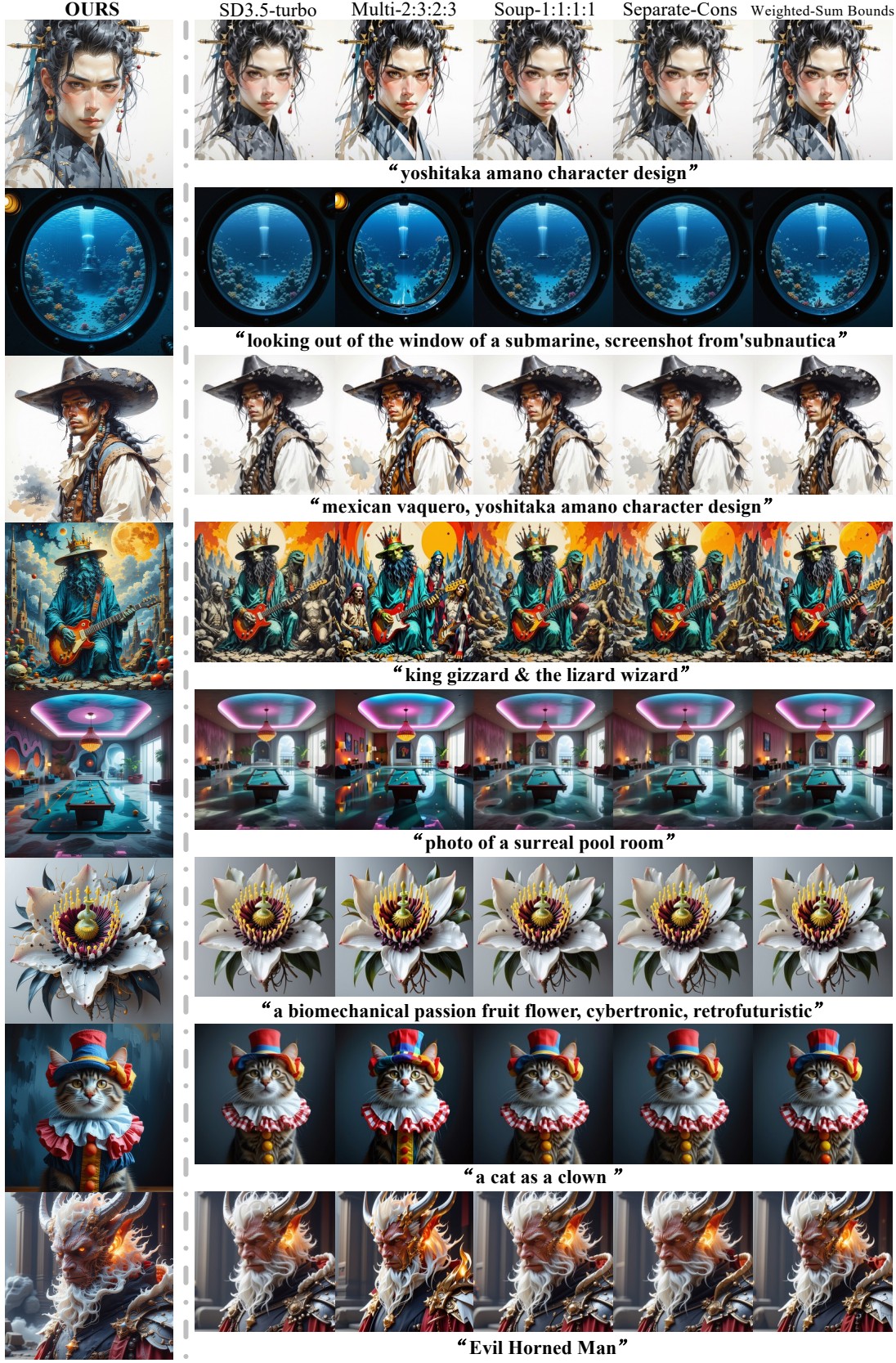

*Figure 12.* Qualitative Comparison of Optimization Results with Multi-Reward Baselines.

