# OpenReview forum: "Pareto-Guided Optimal Transport for Multi-Reward Alignment"
_ICML.cc/2026/Conference — ICML 2026 regular_

### Official Review · Reviewer_zo6L · 2026-03-11

**Soundness:** 3
**Presentation:** 3
**Significance:** 3
**Originality:** 4
**Overall Recommendation:** 5
**Confidence:** 2

**Summary:**

This proposes a novel framework for multi-reward optimization based on optimal transport for text-to-image generation tasks. The proposed method can effectively address hacking and collapse issue by dynamically adjust the optimization objective. The author also proposes two new metrics JDR and JCR to measure reward hacking and model collapse, which is more preferable to the common approach of human verification.

**Compliance With Llm Reviewing Policy:**

Affirmed.

**Final Justification:**

Concerns are fully resolved.

**Key Questions For Authors:**

1. Please include DiffusionNFT baseline, which seems to show that it is possible to imporve all rewards effectively with a properly designed weight-sum baseline.
2. Can the authors provide discussions on the degradation of CLIP scores?  In particular, it has negative win rate (<50%) in Table 3 but higher mean (Table 5). Additional discussions will help readers to better understand these results.
3. What is the statistic significance of the win rate? I imagine they will have higher variance than computing the population mean.  Does higher mean value implies that if we scale up the number of generations per prompt, the expected win-rate should be >50%?

**Limitations:**

yes

**Strengths And Weaknesses:**

Strength:

1. The proposed method employs optimal transport for multi-reward optimization problem, which is a novel paradigm and can inspire additional works.
2. The proposed new metrics seems more reliable and scalable than human verification, which is a significant contribution for reward-based T2I optimization literature.
3. The experiments are comprehensive, including multiple weight combinations for Reward Soup and Joint Optimization.

Weakness:

1. Several alternative baselines should be considered. Such as DiffusionNFT, which showed unformed reward improvements over multiple rewards by 1) design weights based on the statistics of each reward (normalize by mean absolute deviation (MAD)) , as opposed to simple weights like 1:1:4:4, 1:1:1:1 2) 2) Use a training schedule that iteratively optimize different reward combinations in a sequential order.  In general, the biggest concern is that DiffusionNFT's result showed that we can achieve unified improvements over all rewards with properly designed "weight-sum" baseline, while results of this work show degradation in some rewards ( CLIP).

---

> ### Author Rebuttal · Authors · 2026-03-31
>
> We appreciate the reviewer's time and effort in providing such constructive feedback. We address each concern below.
>
> **Q1: DiffusionNFT baseline**
>
> **A1:** Thank you for the suggestion. We have implemented DiffusionNFT on our SD3.5-Turbo backbone with the same four rewards and evaluated on the Parti-Prompts dataset:
>
> **Table 1: DiffusionNFT Results on Parti-Prompts**
> | Model | ICT(%)↑ | HP(%)↑ | CLIP(%)↑ | HPS(%)↑ | JDR₂(%)↑ | JDR₄(%)↑ | JCR₄(%)↓ |
> |---|---|---|---|---|---|---|---|
> | DiffusionNFT | 50.98 | **91.48** | **53.49** | 40.14 | 46.81 | 14.70 | 2.65 |
> | Ours | **56.43** | 85.23 | 43.63 | **61.70** | **47.98** | **17.10** | **2.39** |
>
>
> Although DiffusionNFT achieves strong performance on individual rewards such as HP and CLIP, our method outperforms it on all joint metrics (JDR₂, JDR₄, and JCR₄). The core difference lies in that DiffusionNFT optimizes toward a unified global target across all prompts, whereas our method constructs prompt-specific Pareto frontiers that respect heterogeneous reward upper bounds and leverages optimal transport for optimization, while further detecting and removing weak reward dimensions susceptible to model collapse.
>
>
>
> ---
>
> **Q2: Discussion of CLIP score**
>
> **A2:** Thank you for this careful and perceptive observation. Win rate and mean indeed capture different aspects of performance. Win rate reflects the proportion of samples that improve, while mean represents the overall shift in magnitude. Specifically, in our results, although fewer than half of the samples achieve higher CLIP scores after optimization, those that do improve by a considerable margin, while the majority of the remaining samples experience only relatively minor CLIP score decreases. Consequently, the significant gains from the improved subset are sufficient to offset the small losses from the rest, resulting in a higher overall mean despite a win rate below 50%.
>
> Additionally, CLIP is identified as a weak reward by our VLM agent (Table 1: only 60.30% human preference prediction accuracy, far below ICT's 87.58% and HP's 88.47%) and is intentionally removed during training to prevent reward hacking. This is a deliberate trade-off: the minor CLIP win rate reduction enables substantial gains on strong rewards (ICT: 56.43%, HP: 85.23%) and joint metrics (JDR₂: 47.98%, JDR₄: 17.10%), with genuine quality improvement confirmed by the ~80% human win rate. We hope this explanation addresses the reviewer's concern.
>
>
> ---
>
>
> **Q3: Statistical significance and mean vs. win rate**
>
>
> **A3:** We sincerely thank the reviewer for this rigorous and insightful question, which allows us to further clarify the statistical properties of our experimental results.
>
> The reviewer notes that win rate, as a binomial statistic, tends to have higher variance than the population mean. To ensure rigorous evaluation, we compute 95% confidence intervals (CI) using the Wald binomial proportion method:
>
> $$\text{CI} = \hat{p} \pm z_{0.025} \sqrt{\hat{p}(1-\hat{p})/n},$$
>
> where $n = 1{,}631$ is the number of evaluation prompts. The results are as follows:
>
> | Reward | Win Rate (%) | 95% CI |
> |---|---|---|
> | ICT | 56.43 | [54.02, 58.84] |
> | HP | 85.23 | [83.51, 86.95] |
> | HPS | 61.70 | [59.34, 64.06] |
> | CLIP | 43.63 | [41.22, 46.04] |
>
> As shown, the confidence intervals for ICT, HP, and HPS all lie entirely above 50%, confirming that the improvements are statistically significant despite the higher variance. CLIP lies significantly below 50%, reflecting the inherent trade-off in multi-reward optimization. We will include these confidence intervals in the revision.
>
> Regarding whether a higher mean implies that the win rate would exceed 50% by increasing the number of generations per prompt, this is not necessarily the case. Mean and win rate characterize different aspects of the distribution: mean reflects the magnitude of improvement and can be influenced by a small number of substantially improved samples, whereas win rate measures the proportion of samples that show improvement. In our case, although fewer than half of the samples improve on the CLIP metric, those that do improve by a substantial margin, thereby raising the overall mean. Increasing the number of generations per prompt only reduces estimation variance and cannot change the underlying prompt-wise heterogeneity or reward trade-offs; thus, the win rate would not be expected to change drastically.
>
> This further highlights the limitation of relying solely on per-reward mean or win rate. Our proposed Joint Domination Rate (JDR) is designed to evaluate whether individual samples achieve synergistic improvement across all rewards simultaneously, thereby providing a more faithful measure of multi-reward optimization quality.
>
> We thank the reviewer again for this valuable question, which strengthens the statistical interpretation of our work.

---

> > ### Author Rebuttal · Reviewer_zo6L · 2026-04-03
> >
> > Concerns are fully resolved.

---

> > > ### Author Response · Authors · 2026-04-06
> > >
> > > We sincerely thank the reviewer for confirming that all concerns have been fully resolved. The reviewer's rigorous and insightful questions helped us meaningfully strengthen both the clarity and rigor of our paper. We deeply appreciate the reviewer's time and constructive engagement throughout this process.

---

### Official Review · Reviewer_9HDq · 2026-03-12

**Soundness:** 3
**Presentation:** 2
**Significance:** 3
**Originality:** 2
**Overall Recommendation:** 4
**Confidence:** 3

**Summary:**

The paper demonstrates that using a single objective for multi‑reward text‑to‑image alignment brings reward hacking, and proposes a Pareto‑frontier‑guided optimal transport framework that constructs prompt‑specific Pareto fronts and transports generated samples toward these frontiers. It introduces two complementary optimization strategies, online for strong reward models and offline for weak ones. And this paper defines 2 metrics, Joint Domination Rate and Joint Collapse Rate, to quantify multi‑reward synergy and hacking. Experiments on state‑of‑the‑art diffusion models show obvious JDR gains, near‑80 % human win rate, and substantially reduced reward hacking compared with baseline methods.

**Compliance With Llm Reviewing Policy:**

Affirmed.

**Final Justification:**

The author's feedback has addressed most of the reviewer's concerns and the reviewer chooses to maintain the score.

**Key Questions For Authors:**

Could the author provide more insights in comparing to Parrot and CaPO?

**Limitations:**

The authors are better to provide more discussions about the limitations.

**Strengths And Weaknesses:**

Strengths:

This paper raises the issue of the relationship between multiple rewards and a single prompt.
It proposed a principled, distribution‑aware transport mechanism that adapts to each prompt’s Pareto frontier, producing robust multi‑reward alignment.
It intorduces new metrics (JDR/JCR) and show strong empirical results including human studies that convincingly prove the method works.

Weaknesses:
The paper does not adequately discuss the limitations. It is uncertain about potential failure modes or scalability issues.
The evaluation experiment of the generation backbone is limited to a single model (Stable Diffusion 3.5) and only on Parti‑Prompts, which weakens the generality of the reported results.
The baseline comparison omits related influential works, Parrot(ECCV)[1] and CaPO(CVPR)[2] and this paper should discuss the differences between them more.
Figure captions are short and expanding them to highlight the key takeaways would help.

[1] Parrot: Pareto-optimal Multi-Reward Reinforcement Learning Framework for Text-to-Image Generation
[2] Calibrated Multi-Preference Optimization for Aligning Diffusion Models

---

> ### Author Rebuttal · Authors · 2026-03-31
>
> We thank the reviewer for the constructive feedback and the recognition of our principled framework, new metrics, and strong empirical results. We address each concern below.
>
> ---
>
> **Q1: Comparison with Parrot and CaPO**
>
> **A1:** We thank the reviewer for raising this point. We have already provided an overview of both methods in Appendix A. Here we provide a more detailed comparative analysis below.
>
> **vs. Parrot :** Parrot makes a valuable contribution by introducing Pareto dominance into the sample selection process for multi-reward alignment. It leverages a Prompt Expansion Network to enrich prompt diversity and selects Pareto-dominant sample pairs for reward-weighted RL, effectively improving alignment across multiple objectives. That said, the prompt expansion module introduces additional architectural overhead and may cause semantic drift from the original prompt. Moreover, since alignment operates at the sample-pair selection level, it does not fully exploit distribution-level information. In contrast, our OT-based formulation directly transports the entire batch distribution toward prompt-specific Pareto frontiers, enabling finer-grained and more stable alignment without auxiliary modules. Another key distinction is that Parrot treats all reward models as equally reliable, whereas our framework explicitly distinguishes strong from weak reward models and deploys a VLM-based agent to detect and exclude collapsed rewards, addressing a practical failure mode not considered in Parrot.
>
> **vs. CaPO :** CaPO offers an insightful perspective by recognizing the importance of reward scale calibration within the DPO framework. Its calibration mechanism effectively balances the magnitudes of different reward signals, improving training stability for multi-reward preference optimization. That said, scale calibration addresses reward magnitude imbalance but not the deeper structural issue we identify: different prompts exhibit different feasible reward upper bounds, and a unified global optimization target systematically pushes easy prompts into over-optimization while under-serving hard ones. Our prompt-specific Pareto frontier + OT mapping directly addresses this per-prompt heterogeneity. Beyond this conceptual difference, CaPO requires pre-collected paired preference data, while our framework operates without such paired annotations. Additionally, CaPO does not provide mechanisms for detecting weak reward model collapse, which our framework handles explicitly.
>
>
> ---
>
> **Q2: Limited evaluation (single backbone, single benchmark)**
>
> **A2:** We appreciate this concern. To verify generality, we adopt Flux.1-Schnell as an alternative backbone and evaluate on two additional out-of-distribution prompt benchmarks, EvalMuse[1] and GenAI-Bench[2].
>
> **Table 1: Results on EvalMuse**
> | Model | ICT(%)↑ | HP(%)↑ | CLIP(%)↑ | HPS(%)↑ | JDR₂(%)↑ | JDR₄(%)↑ | JCR₄(%)↓ |
> |---|---|---|---|---|---|---|---|
> | Baseline HPS | 43.22 | 20.60 | 39.69 | 66.33 | 8.04 | 4.02 | 14.07 |
> | Baseline HP | 46.23 | 25.63 | 40.20 | 38.69 | 11.56 | 3.51| 18.09 |
> | Baseline 2:3:2:3 | 45.72 | 19.09 | 46.73 | 44.22 | 7.53 | 2.51 | 17.58|
> | Ours | 42.21 | 17.58 | 45.26 | 62.81 | 9.54 | **4.53** | **13.57** |
>
>
> **Table 2: Results on GenAI-Bench**
> | Model | ICT(%)↑ | HP(%)↑ | CLIP(%)↑ | HPS(%)↑ | JDR₂(%)↑ | JDR₄(%)↑ | JCR₄(%)↓ |
> |---|---|---|---|---|---|---|---|
> | Baseline HPS | 43.44 | 17.06 | 42.37 | 65.75 | 7.81 | 3.50 | 14.25|
> | Baseline HP | 44.56 | 22.37 | 27.00 | 22.37 | 11.31 | 3.06 | 23.31 |
> | Baseline 2:3:2:3 | 47.18 | 24.37 | 49.56 | 42.63 | 12.43| 4.93 | 16.68 |
> | Ours | 47.56 | 23.19 | 50.00 | 63.81 | 11.94 | **5.94** | **11.63** |
>
>
> As shown in Tables 1 and 2, our method consistently achieves the best JDR₄ and the lowest JCR₄ across both benchmarks, confirming its ability to jointly satisfy multiple rewards.
>
>
> [1]Shuhao Han et al., "EvalMuse-40K: A Reliable and Fine-Grained Benchmark with Comprehensive Human Annotations for Text-to-Image Generation Model Evaluation." arXiv preprint arXiv:2412.18150, 2024.
>
> [2] Baiqi Li et al.,"GenAI-Bench: Evaluating and Improving Compositional Text-to-Visual Generation." arXiv preprint arXiv:2406.13743, 2024.
>
>
> ---
>
> **Q3: Discussion of limitations**
>
> **A3:** We thank the reviewer for the careful attention. A key challenge in multi-reward optimization is scalability: our method requires gradient backpropagation through each reward model, so incorporating too many rewards incurs substantial memory overhead. More fundamentally, as shown in our paper, naively adding more rewards can degrade performance, since weak reward models are prone to reward hacking and destabilize training. We will discuss this limitation more explicitly in the revised paper.
>
> ---
>
> **Q4: Short figure captions**
>
> **A4:** We thank the reviewer for this helpful suggestion. In the revised version, we will expand all figure captions to better convey the key takeaways.

---

> > ### Author Rebuttal · Reviewer_9HDq · 2026-04-05
> >
> > Thanks for the author's feedback which addressed most of the reviewer's concerns and the reviewer will maintain the score.

---

> > > ### Author Response · Authors · 2026-04-06
> > >
> > > We thank the reviewer for the positive response and are pleased that our rebuttal resolved most of the concerns. Should any points remain unclear or warrant further discussion, we would be glad to offer additional clarifications. We are grateful for the reviewer's thoughtful feedback, which has been valuable in refining our work.

---

### Official Review · Reviewer_wNud · 2026-03-12

**Soundness:** 2
**Presentation:** 4
**Significance:** 3
**Originality:** 4
**Overall Recommendation:** 4
**Confidence:** 4

**Summary:**

This paper addresses multi-reward reward-hacking by proposing a Pareto Frontier–Guided Optimal Transport framework. The method builds prompt-specific Pareto frontiers and uses distribution-aware optimal transport to map dominated samples toward the frontier, with both online and offline optimization strategies tailored to different reward signal characteristics. The authors introduce two principled metrics—Joint Domination Rate (JDR) and Joint Collapse Rate (JCR)—to quantify multi-reward synergy and reward hacking, and provide comprehensive experiments showing the approach outperforms several baselines.

**Compliance With Llm Reviewing Policy:**

Affirmed.

**Final Justification:**

The authors have provided a detailed response that resolves my concerns. I will upgrade my rating to 4.

**Key Questions For Authors:**

1. How can we ensure the correctness of the prior (the definition of early-stage hacking)?
2. How does it perform on models other than SD3.5 Turbo?
3. Why is single HP the strongest baseline—why do reward-system approaches fail to outperform it?

**Limitations:**

1. It is difficult to scale to GRPO, which requires evaluating multiple samples per iteration and has slower convergence efficiency, resulting in very high costs.
5. Experiments show that a strong reward is effective (HP outperforms the reward system), but it remains an open question whether a more powerful unified multi-dimensional reward (e.g., UnifiedReward [1]) would be more useful than the reward system in this paper.

[1] Wang, Y., Li, Z., Zang, Y., Wang, C., Lu, Q., Jin, C. and Wang, J., Unified Multimodal Chain-of-Thought Reward Model through Reinforcement Fine-Tuning. In The Thirty-ninth Annual Conference on Neural Information Processing Systems.

**Strengths And Weaknesses:**

Strengths：
1. The paper studies reward hacking—particularly proposing the multi-reward reward-hacking problem—and is novel; JDR and JCR provide useful insights for related research.
2. The paper’s experiments outperform several baselines.

Weaknesses：
1. Heavy reliance on a VLM-based agent introduces high costs and reliability concerns：
(a) Additional cost and instability: Open-source or relatively weaker VLMs (e.g., Qwen3-VL, GPT-4o) are known to suffer from severe hallucination issues. In practice, minor variations in system prompts or image ordering can lead to inconsistent outputs. Using closed-source VLMs (Gemini 2.5 Pro, GPT-5) to improve robustness would introduce non-trivial monetary costs and reduce accessibility. Furthermore, the paper only reports VLM agent accuracy on 200 sampled cases, but this evaluation lacks sufficient details for reproducibility. It is unclear whether these 200 samples are drawn from early training iterations (which are typically more challenging due to subtle differences). The evaluation protocol is under-specified: the ground-truth label format and exact evaluation metrics are not described. Without these details, the reported accuracy is difficult to verify or reproduce.
(b) Generalization limitations: To mitigate hallucination, the paper uses early-iteration data as prior knowledge for the VLM agent (Line 251). However, this design may limit generalization across base models. SD 3.5 Turbo is a small, fast model that typically produces images with weaker composition and lower visual complexity than stronger models. It remains unclear whether this prior remains valid when adapting the framework to other base models. If the prior becomes mismatched, the reported 200-sample accuracy could degrade substantially.

2. The paper argues that weak reward models tend to capture shortcuts, which can lead to reward hacking. However, this claim is largely presented as an empirical hypothesis rather than a theoretically justified result.

3. The algorithm implementation is complex: it requires pre-computing a candidate set of samples, extracting the Pareto frontier, defining a prior for early-iteration hacking (Line 251), and selecting a VLM agent.

---

> ### Author Rebuttal · Authors · 2026-03-31
>
> We thank the reviewer for their rigorous review and helpful suggestions. We address each concern below.
>
> ---
>
> **Q1: Cost, Reliability, and Generalization**
>
>
> **A1(a): Cost and Stability.** As noted in Section 4.1, the VLM agent is invoked only once every 100 steps at \$0.015 per detection, merely 0.4% of total training cost. Open-source VLMs are sufficiently reliable: Qwen3-VL-32B-Thinking (87.5%) and GLM-4.5V (84.0%) perform comparably to GPT-4o (90.5%), confirming our approach does not hinge on closed-source models. Our evaluation samples span different training stages and predominantly cover early iterations, further validating the agent's cost-effectiveness and reliability.
>
> For Table 4, ground-truth labels were annotated by human experts selecting from [CLIP, HPS, ICT, HP, None] to indicate which reward had collapsed. The VLM agent outputs in the same format, and accuracy measures agreement with these human labels. The 200 samples span diverse baselines and multiple collapsed checkpoints, emphasizing early-stage collapse. As Figure 10 shows, each reward model exhibits highly consistent collapse patterns across checkpoints, making reliable detection feasible.
>
> **A1(b): Generalization.** We thank the reviewer for this important point. Figure 10 shows that these patterns hold consistently across optimization methods, indicating they are inherent to the reward signal itself. When the base model differs significantly, separately sampling each reward to re-acquire its collapse patterns is a modest one-time cost. Without it, naively combining multiple rewards wastes compute and lacks interpretability.
>
> ---
>
> **Q2: Theoretical Justification**
>
> **A2:** We appreciate the reviewer for raising this concern. Under a fixed KL constraint, proxy-gold reward divergence accelerates with increasing reward model misspecification [1]. Given the accuracy gap in main paper Table 1 (HP 88.47% vs. CLIP 60.30%), the two rewards exhibit substantially different approximation errors, leading to qualitatively different optimization behaviors. Moreover, optimization pressure drives the policy out-of-distribution, where approximation errors are amplified and lower-accuracy rewards become susceptible to shortcut exploitation [2]. We further hypothesize that in multi-reward settings, this effect concentrates on the weakest reward dimension, resulting in performance collapse along that dimension.
>
> [1] Gao et al., "Scaling Laws for Reward Model Overoptimization," *ICML*, 2023.
>
> [2] Casper et al., "Open Problems and Fundamental Limitations of Reinforcement Learning from Human Feedback," *TMLR*, 2023.
>
>
>
> ---
> **Q3: Algorithm Complexity**
>
> **A3:** Although our framework has multiple components, the actual overhead is minimal: Pareto precomputation is a one-time cost, and per-iteration overhead is only ~3%. The ablation study confirms each component contributes consistent gains, indicating they are necessary and effective.
>
> ---
>
> **Q4: How to ensure correctness of the early-stage hacking prior?**
>
>
> **A4:** Each reward model exhibits characteristic collapse patterns that remain consistent across optimization methods (Appendix E, Figure 10). This consistency ensures priors generalize reliably, as validated by 84–90.5% agreement accuracy on diverse samples.
>
> ---
>
> **Q5: Why is single HP the strongest baseline?**
>
>
> **A5:** HP is the most accurate reward model (acc 88.47%, Table 1) , making it naturally resistant to reward hacking. Existing multi-reward methods fail to properly handle weaker reward signals , overlooking the reward hacking issue introduced by weak rewards, which ultimately degrades overall performance. In contrast, our method actively removes weak rewards via the agent and leverages OT to guide the optimization direction, achieving 47.98% JDR₂ and 17.10% JDR₄ (vs. 36.15% and 13.73% for single HP), validating the synergistic gains of our framework across multiple dimensions.
>
> ---
>
> **Q6: Performance on other base models? Would UnifiedReward be more effective?**
>
> **A6:**  Our method is directly transferable to other base models, and we provide supplementary experiments on Flux.1-Schnell and using UnifiedReward for optimization:
>
> **Table 1: Results on Parti-Prompt**
> | Model | ICT(%)↑ | HP(%)↑ | CLIP(%)↑ | HPS(%)↑ | JDR₂(%)↑ | JDR₄(%)↑ | JCR₄(%)↓ |
> |---|---|---|---|---|---|---|---|
> | Baseline HPS | 44.79 | 17.22 | 39.95 | 66.36 | 8.57 | 3.67 | 13.60 |
> | Baseline HP | 42.58 | 23.83 | 43.50 | 30.63 | 10.29 | 3.43 | 22.30 |
> | Baseline 2:3:2:3 | 44.79 | 22.48 | 48.46 | 39.03 | 10.11 | 3.61 | 17.34 |
> | UnifiedReward | 43.56 | 13.54 | 43.68 | 46.20 | 6.68 | 2.57 | 19.12 |
> | Ours | 49.51 | 19.12 | 46.14 | 57.53 | 10.12 | **4.23** | **13.11** |
>
> Results confirm cross-model generalization of our method. Moreover, even a powerful unified multi-dimensional reward model like UnifiedReward still suffers from trade-offs and reward hacking under weighted-sum optimization, further highlighting the advantage of our framework.

---

> > ### Author Rebuttal · Reviewer_wNud · 2026-04-01
> >
> > Thank you for the authors’ response. I appreciate the clarification provided.
> > However, I still have several substantive concerns that, in my view, remain insufficiently addressed.
> >
> > (1) Generalization concerns remain unresolved. The response still does not provide additional training results using other VLM backbones such as QwenVL or GLM-4.5. As a result, the current evidence still mainly relies on the accuracy numbers reported in Table 4, which by itself is not yet sufficient to support broad generalization claims. In addition, my previous question regarding the evaluation statistics and statistical reliability of the Table 4 results does not appear to have been directly addressed.
> >
> > (2) Regarding generalization on the model side, the presented evidence is still limited to a small-scale distilled model. This leaves open the question of whether the proposed method transfers equally well to stronger or larger base models.
> >
> > (3) The reliance on manually defined hacking patterns remains a concern. Based on the response, the method still appears to require manual definition of different types of reward signals or early-collapse patterns to evaluate VLM accuracy and guide training. The paper argues that detecting collapse as early as possible is particularly important, since failures that occur later are often harder to correct. However, many early failures involve pixel differences too subtle for reliable identification. This makes the manual specification of VLM-specific failure criteria and reward-hacking rules difficult to justify, and arguably no easier to interpret or generalize than directly tuning a reward soup.

---

> > > ### Author Response · Authors · 2026-04-05
> > >
> > > We thank the reviewer for the thoughtful follow-up and address each remaining concern below.
> > >
> > > ---
> > >
> > > **Concern (1): VLM Agent Backbone Generalization**
> > >
> > > **A1(a):**  We conducted complete end-to-end training with Qwen3-VL-32B-Thinking replacing GPT-4o, achieving consistent performance, confirming that current open-source VLMs can fully serve as the decision agent.
> > >
> > >
> > > | Agent | ICT↑ | HP↑ | CLIP↑ | HPS↑ | $JDR_2$↑ | $JDR_4$↑ | $JCR_4$↓ |
> > > | :--- | :---: | :---: | :---: | :---: | :---: | :---: | :---: |
> > > | **Qwen3-VL-32B/T** | 56.00 | **88.11** | 41.91 | 57.97 | **49.30** | 16.67 | **2.21** |
> > > | **GPT-4o** | **56.43** | 85.23 | **43.63** | **61.70** | 47.98 | **17.10** | 2.39 |
> > >
> > > *All values are reported in %.*
> > >
> > > **A1(b):** We elaborate on the statistical reliability of Table 4 as follows.
> > >
> > > **Sample construction and coverage.** The 200 evaluation samples are systematically constructed rather than randomly drawn from a single setting. They span four reward models (CLIP, HPS, ICT, HP), multiple baseline optimization methods (single-reward, multi-reward weighted, reward soup), and multiple training checkpoints, with 80% focused on early-stage collapse, the most challenging detection scenario, and the remaining 20% covering mid-to-late stages.
> > >
> > >
> > > **Statistical confidence.** We compute 95% Wald binomial confidence intervals ($n=200$) as $\text{CI} = \hat{p} \pm z_{0.025} \sqrt{\hat{p}(1-\hat{p})/n}$, yielding *GPT-4o 90.5% [86.4, 94.6]*, *Qwen3-VL-32B/T 87.5% [82.9, 92.1]*, and *GLM-4.5V 84.0% [78.9, 89.1]*. The narrow intervals (±4–5%) and substantial cross-VLM overlap confirm statistical stability and no significant gap between closed- and open-source models.
> > >
> > >
> > > ---
> > >
> > > **Concern 2: Base Model Scalability**
> > >
> > > **A2:** We thank the reviewer and provide supplementary analysis on base model generalization below.
> > >
> > > We respectfully note that Flux.1-Schnell (12B), included in our previous rebuttal, already differs fundamentally from SD 3.5 Turbo (8.1B) in both architecture and scale, providing strong evidence of generalization. To further address the concern regarding distilled models, we conduct additional experiments on SD 3.5 Medium, a non-distilled base model, where our method achieves the best joint metrics, confirming cross-architecture generalization.
> > >
> > > | Model | ICT↑ | HP↑ | CLIP↑ | HPS↑ | $JDR_2$↑ | $JDR_4$↑ | $JCR_4$↓ |
> > > | :--- | :---: | :---: | :---: | :---: | :---: | :---: | :---: |
> > > | Baseline-hp | 49.33 | 13.35 | 39.95 | 79.41 | 7.23 | 2.69 | 6.98 |
> > > | Baseline-2323 | 43.01 | 11.71 | 50.67 | 40.80 | 4.78 | 1.89 | 16.72 |
> > > | Ours | 49.38 | 14.09 | 49.52 | 80.15 | **7.41** | **3.43** | **6.23** |
> > >
> > > *All values are reported in %.*
> > >
> > > ---
> > >
> > > **Concern 3: Manual Dependency of Collapse Detection**
> > >
> > > **A3(a): Early-stage detection reliability.** The reviewer's concern regarding early-stage "subtle differences" in fact points to a critical trade-off: premature intervention would suppress the positive effect the Reward Model is still exerting. If the "early-stage failures" envisioned by the reviewer are pixel-level indistinguishable from normal images, this precisely indicates that the RM is still providing effective guidance and Reward Hacking has not yet occurred. By the accepted definition [1][2], Reward Hacking is characterized by rising reward scores coupled with perceptible quality degradation below that of normal images. Hence, pixel-level differences warrant no intervention, and once intervention is warranted, artifacts must already be structurally detectable. Figure 10 confirms that collapse patterns are reward-model-specific and structurally consistent, providing a reliable detection basis. Our agent detects every 100 steps, a granularity sufficient to capture such manifested artifacts; upon detection, the corresponding reward is removed and the model reverts to a healthy state. Three architecturally diverse VLMs all achieve 84–90.5% accuracy, empirically validating the feasibility and robustness of detection at this stage.
> > >
> > > **A3(b): Manual effort vs. reward soup.** Human judgment is an unavoidable ground truth for any hacking detection. Early stopping relies on human judgment [3], but is neither transparent nor reproducible. Our method structuralizes this judgment: a small set of annotated samples enables VLMs to automatically learn degradation patterns. Moreover, collapse characteristics of a given reward model are consistent across base models (e.g., over-saturation for HPS, blurring for CLIP), making pattern definition a one-time investment. Reward soup, by contrast, can only adjust fusion weights and cannot repair an already-collapsed signal source. Even after exhaustive weight search, its best $JDR_2$ is only 27.02%, while ours reaches 47.98%.
> > >
> > > [1] Clark et al., "Directly Fine-Tuning Diffusion Models on Differentiable Rewards," ICLR, 2024.
> > >
> > > [2] Zhang et al., "Large-scale Reinforcement Learning for Diffusion Models," arXiv, 2024.
> > >
> > > [3] Li et al., "TextCraftor: Your Text Encoder Can be Image Quality Controller," CVPR, 2024.

---

### Official Review · Reviewer_cCZd · 2026-03-13

**Soundness:** 3
**Presentation:** 3
**Significance:** 3
**Originality:** 3
**Overall Recommendation:** 4
**Confidence:** 4

**Summary:**

This paper tackles the issue of "reward hacking" in the RL for text-to-image (T2I) generation models, particularly multi-reward alignment. The authors identify a limitation in existing approaches, which rely on global reward targets that disregard the heterogeneous upper bounds of rewards associated with different prompts. This misalignment often leads to the reward hacking problem.

To address this issue, the authors propose a Pareto-Frontier-Guided Optimal Transport framework, which utilizes prompt-specific Pareto frontiers as optimization targets, where the generated sample distributions are mapped to these frontiers using Optimal Transport (OT) techniques. Furthermore, the paper introduces a VLM-based decision agent designed to prune "weak" reward models prone to inducing collapse. The authors also establish two novel evaluation metrics—Joint Domination Rate (JDR) and Joint Collapse Rate (JCR)—to better analyze and quantify the outcomes of their framework.

**Compliance With Llm Reviewing Policy:**

Affirmed.

**Final Justification:**

The authors have addressed most of my questions. My remaining concern is that in reward hacking scenarios, it is common for one reward model to assign a very high score while another assigns a very low score. Would this situation cause the proposed JDR and JCR metrics to become ineffective? Overall, I maintain my previous score: weak accept.

**Key Questions For Authors:**

1. Have the authors considered a fundamental limitation: what happens if two rewards are inherently incompatible, making it impossible to achieve their optimal values simultaneously (e.g., extreme stylization vs. strict text alignment)?

**Limitations:**

Yes. The authors have adequately discussed the limitations in Sec. C.4.

**Strengths And Weaknesses:**

Strengths

1. The paper provides a formal analysis demonstrating that unified global targets inevitably cause reward hacking under heterogeneous prompt-wise reward bounds.
2. Aligning distributions with the Pareto frontier is technically sound and avoids the manual weight-tuning issues inherent in scalarization methods.
3. The proposed metrics, JDR and JCR, address the limitations of mean-based reward scores, offering a principled way to evaluate multi-objective synergy and failure modes.

Weaknesses

1. The proposed pipeline appears overly complex, encompassing offline training, online strategies, and a VLM-based decision mechanism. To enhance clarity, it would be beneficial to provide a well-structured algorithm outlining the process.
2. The proposed metrics, JDR and JCR, appear promising. However, additional evidence, such as experimental results or further analysis, is needed to convincingly demonstrate their effectiveness.
3. Table 4 presents the ablation study on different VLMs. However, the results show that the VLM-based agent is highly dependent on the choice of VLM. For instance, when switching from GPT-4o to GLM-4.5V, the accuracy drops by 6.5%. This suggests that the robustness of the proposed method is relatively weak. Additionally, it raises the question of whether the selection of VLM should be adapted to different reward models to ensure better compatibility.

---

> ### Author Rebuttal · Authors · 2026-03-31
>
> We sincerely appreciate the reviewer's careful reading and valuable feedback. We address each concern below.
>
> ---
>
> **Q1: Pipeline Complexity — Need for a Structured Algorithm**
>
> **A1:** We sincerely thank the reviewer for the valuable suggestion. We agree that presenting the full pipeline as a unified algorithm would improve clarity. In the revised version, we will include a formal algorithm box (Algorithm 1) to clearly present the complete training procedure. For completeness, we provide the pseudocode below.
>
>
>
> **Algorithm 1: Pareto-Frontier-Guided Optimal Transport**
>
> **Input:** T2I model $G_\theta$, prompts $\mathcal{P}$, reward models $\mathcal{S} = \{R^k\}_{k=1}^K$, VLM agent $\mathcal{A}$
>
>
>
> **Stage 1: Offline Pareto Frontier Construction**
>
> 1: **for** each prompt $p_i \in \mathcal{P}$ **do**
>
> 2: &emsp; Generate $M$ images, compute reward vectors, extract Pareto frontier $\mathcal{R}^{\text{front}}(p_i)$
>
> 3: **end for**
>
>
>
> **Stage 2: Training Loop**
>
> 4: strategy $\leftarrow$ OFFLINE
>
> 5: **for** each training step **do**
>
> 6: &emsp; Sample prompt batch $\mathcal{B}$, generate images, compute rewards
>
> 7: &emsp; **if** strategy = OFFLINE **then**
>
> 8: &emsp;&emsp; Set target $\nu_i \leftarrow \mathcal{R}^{\text{front}}(p_i)$ from precomputed frontiers
>
> 9: &emsp; **else** (ONLINE)
>
> 10: &emsp;&emsp; Set target $\nu_i \leftarrow$ Pareto-dominating vectors from current batch
>
> 11: &emsp; **end if**
>
> 12: &emsp; Solve OT via Sinkhorn, compute loss, update $\theta$
>
> 13: &emsp; **// Agent checkpoint (every $T_c$ steps)**
>
> 14: &emsp; **if** $\mathcal{A}$ detects reward hacking **then** remove collapsed $R^k$ from $\mathcal{S}$, rollback $\theta$
>
> 15: &emsp; **if** all remaining $R^k$ are strong and stable **then** strategy $\leftarrow$ ONLINE
>
> 16: **end for**
>
> ---
>
>
> **Q2: Additional Evidence for JDR and JCR Effectiveness**
>
> **A2:** The effectiveness of JDR and JCR is grounded in a key property: reward models generally agree on high-quality images but diverge on degraded ones, since different rewards exhibit distinct collapse patterns. Shortcut artifacts that inflate one reward are often detected as anomalies by others. Therefore, JDR, which requires all rewards to improve, can only be high when the optimization is genuinely correct and free of collapse, serving as both a performance metric and a reward hacking detector. In contrast, JCR, which measures when all rewards decline, captures global degradation that single-reward metrics cannot reveal and thus provides a more rigorous measure of collapse.
>
> Detailed empirical analysis further supports this. (1) Multi-2:3:2:3 achieves an 86.03% HPS win rate, yet its JDR₄ is only 13.42%, showing that most samples fail to improve across all rewards, a failure not visible from individual win rates. (2) The single-reward ICT baseline achieves a 56.99% win rate but has JCR₄ = 10.17%, indicating that more than 10% of samples suffer from hidden global degradation. (3) Our approximately 80% human win rate (Table 6) strongly correlates with leading JDR scores, confirming that JDR captures the quality dimensions humans care about more effectively than mean-based metrics.
>
> ---
>
> **Q3: VLM Agent Robustness**
>
> **A3:** Thank you for this question. Even our lowest-performing VLM (GLM-4.5V) already achieves 84.0% accuracy, and in practice we invoke the agent multiple times per checkpoint with majority voting, further improving reliability. The agent also operates conservatively: false positives only trigger a safe rollback costing ~100 steps, posing minimal risk. Regarding VLM-reward compatibility, all three VLMs converge to identical pruning decisions (removing HPS at step 200 and CLIP at step 600). This is because different reward models produce structurally distinct collapse patterns (e.g., HPS: over-saturation; CLIP: pixelation), making detection straightforward for any capable VLM. We agree that as reward models evolve, leveraging diverse VLMs or ensemble strategies to mitigate potential blind spots is a valuable direction worth exploring. We appreciate the reviewer for raising this point.
>
> ---
>
> **Q4: Fundamentally Incompatible Rewards**
>
> **A4:** When two rewards are fundamentally incompatible, a natural trade-off emerges on the Pareto frontier. The point where both rewards simultaneously reach their optima is the **ideal point**, which is mathematically well-defined but generally unattainable in practice. Crucially, our method does not force optimization toward this unrealistic ideal. Instead, we use optimal transport to map each dominated sample to its nearest point on the Pareto frontier, preserving the inherent trade-off structure rather than collapsing all samples toward a single target. If all candidates already lie on the frontier for a given prompt, no transport is applied, naturally indicating convergence. This ensures our framework gracefully handles reward incompatibility without overstepping feasible bounds.

---

> > ### Author Rebuttal · Reviewer_cCZd · 2026-04-03
> >
> > The authors have addressed most of my questions. My remaining concern is that in reward hacking scenarios, it is common for one reward model to assign a very high score while another assigns a very low score. Would this situation cause the proposed JDR and JCR metrics to become ineffective? Overall, I maintain my previous score: weak accept.

---

> > > ### Author Response · Authors · 2026-04-05
> > >
> > > **Concern 1: Effectiveness of JDR/JCR under Extreme Score Divergence**
> > >
> > > **A1:** We thank the reviewer for the recognition and this follow-up question.
> > >
> > > The scenario where one reward model assigns extremely high scores while another assigns extremely low scores is precisely the typical manifestation of reward hacking, exposing the fundamental limitation of existing multi-reward evaluation metrics. JDR and JCR are designed exactly to address this.
> > >
> > > **Limitations of existing metrics.** Reward averaging masks cross-dimensional divergence through cancellation of high and low scores, making the overall mean appear normal while optimization is in fact severely imbalanced. Per-reward win rate only reflects local changes along individual dimensions: the hacked reward exhibits a spuriously high win rate, while degradation in other rewards is scattered across independent metrics, unable to provide a unified sample-level diagnosis.
> > >
> > > **How JDR and JCR address this.** JDR requires all reward dimensions to simultaneously outperform the baseline on each sample; any single-dimension degradation disqualifies that sample. Therefore, over-optimization of a single reward cannot translate into a higher JDR; only stable improvement across all dimensions can. JCR measures the proportion of samples where all rewards degrade simultaneously, capturing global collapse that averaging fails to reveal. Together, they provide complementary diagnostics from the perspectives of *"joint improvement"* and *"joint degradation."*
> > >
> > >  Consequently, under extreme divergence, JDR will not be misled like averaging nor spuriously inflated like per-reward win rate, but instead faithfully reflects the inconsistency of multi-dimensional optimization. **Extreme score divergence is precisely the scenario where JDR/JCR are most discriminative.**
> > >
> > > ---
> > >
> > > We hope this response addresses the concern. Should there be any remaining questions, we are happy to provide further clarification or additional experiments.

---

### Decision · Program_Chairs · 2026-04-30

**Decision:**

Accept (regular)

**Comment:**

1. Multiple novel contributions : transport of samples to the Pareto frontier, new metrics, and VLM based pruning of weak rewards . The first 2 could easily be influential on the field.

2. Addresses an important challenge for preference-based optimization:  reward hacking and how to optimize for multiple objectives.

3. Empirical results have not entirely satisfied all reviewers, since they could be more comprehensive e.g. more baselines and complex prompts beyond Parti-prompts. This penalizes the score to less than strong accept. But otherwise a good novel contribution likely to spur new research.